# Metallopolymer strategy to explore hypoxic active narrow-bandgap photosensitizers for effective cancer photodynamic therapy

Zhao Zhang [1], Zixiang Wei[1], Jintong Guo[1], Jinxiao Lyu[1], Bingzhe Wang[2], Gang Wang[2], Chunfei Wang[1], Liqiang Zhou [1,3], Zhen Yuan [1,3], Guichuan Xing [2], Changfeng Wu [4] & Xuanjun Zhang [1,3] ✉

Practical photodynamic therapy calls for high-performance, less $O_2$-dependent, long-wavelength-light-activated photosensitizers to suit the hypoxic tumor microenvironment. Iridium-based photosensitizers exhibit excellent photocatalytic performance, but the in vivo applications are hindered by conventional $O_2$-dependent Type-II photochemistry and poor absorption. Here we show a general metallopolymerization strategy for engineering iridium complexes exhibiting Type-I photochemistry and enhancing absorption intensity in the blue to near-infrared region. Reactive oxygen species generation of metallopolymer **Ir-P1**, where the iridium atom is covalently coupled to the polymer backbone, is over 80 times higher than that of its mother polymer without iridium under 680 nm irradiation. This strategy also works effectively when the iridium atom is directly included (**Ir-P2**) in the polymer backbones, exhibiting wide generality. The metallopolymer nanoparticles exhibiting efficient $O_2^{\bullet-}$ generation are conjugated with integrin $\alpha v\beta 3$ binding cRGD to achieve targeted photodynamic therapy.

Photosensitive materials open a new era for a noninvasive therapeutic form of photodynamic therapy (PDT) in a highly spatially and temporally controlled manner[1,2]. In the PDT process, photosensitizers are excited to generate toxic reactive oxygen species (ROS), including superoxide radicals ($O_2^{\bullet-}$), hydroxide radicals ($^{\bullet}OH$), and hydrogen peroxide ($H_2O_2$) via electron transfer (Type-I) and singlet oxygen ($^1O_2$) via energy transfer (Type-II)[3,4]. Most organic photosensitizers exhibit $O_2$-demanding Type-II ROS generation, whereas the hypoxic microenvironment is a typical feature of tumors. To circumvent this obstacle, light-triggered photochemotherapy[5,6], synergistic photoimmunotherap[7], and mitochondria respiration inhibition strategy[8] have been applied with PDT response enhanced. $O_2$ self-sufficient strategies have been successful[9,10], but the quantity of exogenous $O_2$ draws the invisible limit for the working drug concentration.

There is a strong need for Type-I photosensitizers. Current studies show that energy level tuning[11,12], strong electron-donating materials construction[13,14], biotinylation[15], and supramolecular assembly[16–18] could favor Type-I ROS generation. Despite these excellent developments, the Type-I ROS generation is like a bonus property of photosensitizers with its uncertainty to theoretically predict actual ROS type for a certain structure. Consequently, engineering the existing Type-I photosensitizers[19,20] is of great value as discovering new ones for application-oriented research.

Organic iridium(III) complexes are a type of photosensitizers with excellent photocatalytic properties since the bountiful orbits of iridium atoms promote the spin-orbit-coupling, lead to a higher probability of undergoing intersystem crossing, and improve ROS generation[21,22]. However, a major drawback of this type is their

[1]Cancer Centre and Centre of Reproduction, Development and Aging, Faculty of Health Sciences, University of Macau, Macau SAR 999078, China. [2]Institute of Applied Physics and Materials Engineering, University of Macau, Macau SAR 999078, China. [3]MOE Frontiers Science Centre for Precision Oncology, University of Macau, Macau SAR 999078, China. [4]Department of Biomedical Engineering, Southern University of Science and Technology, Shenzhen, Guangdong 518055, China. ✉e-mail: xuanjunzhang@um.edu.mo

short-wavelength absorption and low absorption coefficient. Coordination with long-wavelength absorption chromophore[23–25], donor-acceptor substructure ligand[26,27], or multiphoton-absorption ligand[28–30] can mitigate this problem. Another limitation is their normally reported Type-II photosensitization[31–33]. It is difficult to integrate the two characteristics of excellent absorption and Type-I ROS generation in a simple and universal way into one molecule. For those limitedly reported Type-I Ir(III) photosensitizers, most of them mainly absorb light in 400 - 500 nm ranges[34–40], and even fewer of them can be excited by bio-friendly lasers[41].

Herein, we report a narrow-bandgap metallopolymerization strategy to prepare high-performance iridium-based metallopolymers with high $O_2^{\cdot-}$ generation in the presence of a 680 nm laser. Type-I Ir(III) unit may ensure corresponding pathways and polymerization can improve absorption properties[42,43] and further ROS generation efficacy[44,45]. First, the Ir(III) complexes with Type-I ROS generation properties are selected and secondly, the candidate complexes are chemically modified to enable polymerization with a suitable bridging unit to form metallopolymers. Further configuration change of Ir(III) monomers does not hamper the $O_2^{\cdot-}$ generation and superior absorption properties. Our finding strongly indicates that the narrow-bandgap metallopolymerization strategy would be a facile but efficient method to acquire high-performance photosensitizers with Type-I ROS generation and intensive long-wavelength light absorption, which not only turns stone into gold but also kills two birds with one stone.

Several quinazoline-based Ir(III) complexes are synthesized and **Ir1** exhibits mixed Type-I and -II photochemical pathways in the presence of white light irradiation. Due to the short and weak absorption, further engineering of this molecule is necessary. The narrow-bandgap

polymer (**PPy-DPP**) without a catalytic iridium center shows an advantage in only strong long-wavelength absorption but a very poor ROS generation efficiency. **Ir1** is introduced into **PPy-DPP** to exploit the individual features of both poor-performance materials. As shown in Fig. 1, the hybridization of the molecular orbit of the Ir(III) molecule and bridging diketopyrrolopyrrole (**DPP**) unit leads to the absorption of metallopolymer (**Ir-P1**) extended to the deep-red region with significantly enhanced intensity. Upon 680 nm laser irradiation, **Ir-P1** harvests energy and generates ROS efficiently, even better than that for Ce6 (Chlorin e6) in dichloromethane or tetrahydrofuran. This efficiency of ROS production is more than 80-fold higher than that of **PPy-DPP** without an iridium center. Importantly, metallopolymer still maintains crucial Type-I photochemistry. The metallopolymerization strategy is applied to **Ir-P2** where the iridium atom is directly included in the backbone. The iridium atom is efficiently involved in the elongation of the polymer's absorption and enhancement of ROS generation. The corresponding metallopolymer nanoparticles (MPdot) **MPdot1** and **MPdot2** exhibit very low dark cytotoxicity but can be activated by a 680 nm laser to show excellent therapeutic effects in vitro and in vivo. This strategy also works well for Type-I ROS-generating metallopolymers with absorptions in blue, green, and NIR regions. These findings indicate that our strategy is effective in the exploration of Type-I metallopolymers by a combination of iridium complexes and semiconducting polymers that show low performance when used individually or mixed physically. This narrow-bandgap metallopolymer strategy can fully exploit the toy brick feature of co-polymerization and is anticipated to find broad use in the exploration of high-performance photosensitizers with Type-I photochemistry and strong absorption in long-wavelength regions.

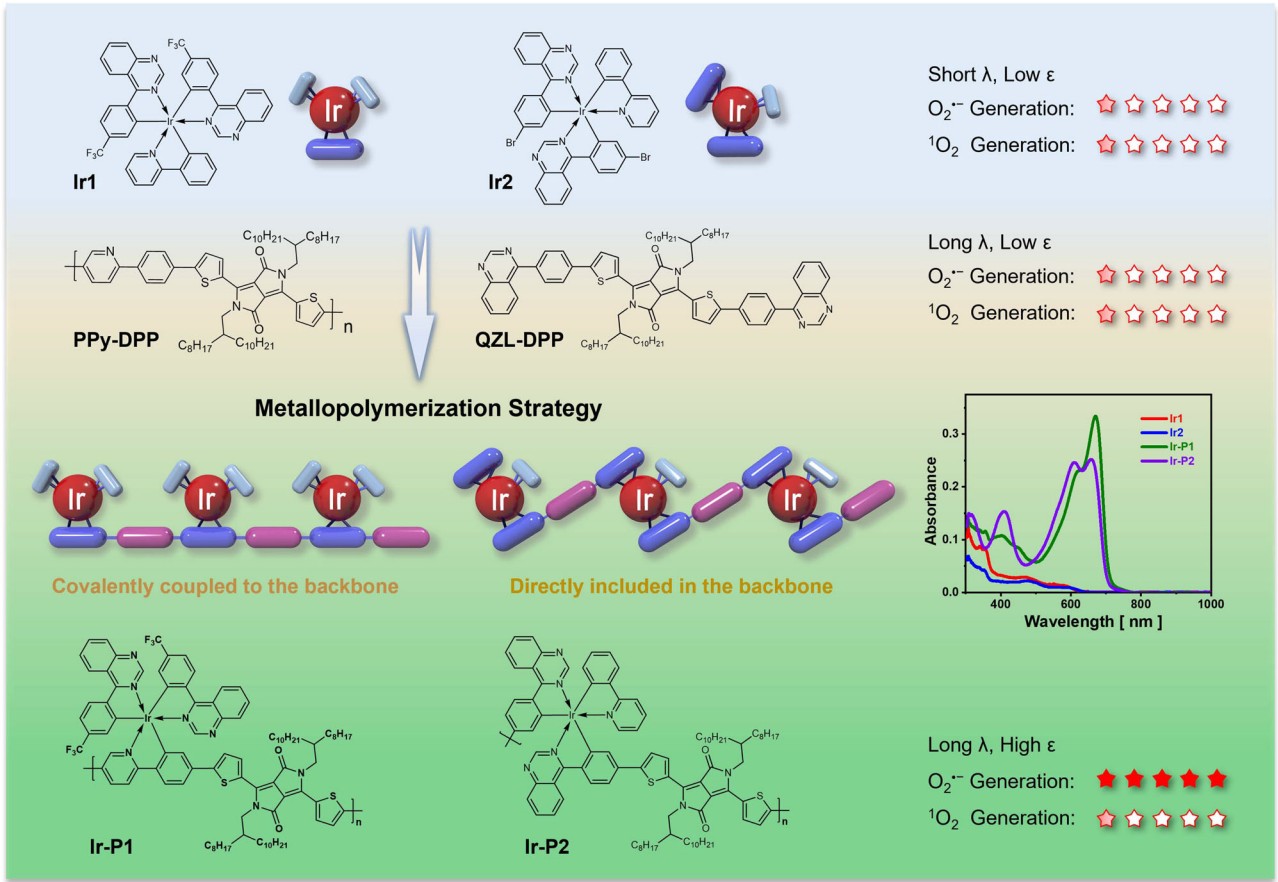

**Fig. 1 | Illustration of the metallopolymerization strategy.** Metallopolymerization to explore high-performance Type-I photosensitizers with strong absorption in the long-wavelength region.

## Results and discussion

### Design and synthesis

The monomers were synthesized according to the previously reported methods with minor modifications in a relatively high yield[46–48], as shown in Supplementary Fig. 1. The compound **Ir-N^N** with bipyridine acts as the auxiliary ligand and exhibits common Type-II ROS generation. However, surprisingly, the replacement of the third ligand with phenylpyridine (**Ir1**) leads to mixed types of photochemical pathways (both Type-I and -II), as shown in Supplementary Fig. 2a, b. A similar phenomenon was also observed in Ru(II) complexes and explained by that the electron-donating cyclometalated C^N-ligands can elevate the energy level of the dπ(Ru) orbital and make the oxidation potential cathodic shift, providing a possibility for Type-I photosensitizers by electron transfer[13,49]. This Type-I photochemical pathway cannot be ignored, and we further engineered **Ir1** to address its problematic absorption and low ROS generation. The di-bromine derivatives **Ir1** and **Ir2** are synthesized and readily available for cross-coupling with stannly or boronic reagents. Many exquisite spectral tunings or other functionalization can be realized efficiently due to this toy brick feature of polymerization. The two metallopolymers studied in this work not only present successful examples but also may lead to the merging of a new series of metallopolymeric photosensitizers. The polymer weight and polydispersity index (for **Ir-P1**, Mn is 26.70 kDa with a PDI of 2.55; and for **Ir-P2**, Mn is 10.92 kDa with a PDI of 1.76, respectively) have been summarized in Supplementary Table 1. Inductively coupled plasma-optical emission spectrometry of these two polymers helps to confirm the successful incorporation of iridium atoms in metallopolymers. Combined with the elemental analysis of C, H, N, and S in each polymer, it can be concluded that the polymers are formed by iridium monomers and **DPP** units with a roughly 1:1 ratio.

### Photophysical and photochemical properties

The photophysical properties of the monomers, organic polymers, and metallopolymers were studied first (Fig. 2a, b). The absorption of **Ir1** and **Ir2** show featured absorption of Ir(III) complexes that hardly extend to a wavelength of 600 nm with low absorption intensity, originating from the ³MLCT (metal-ligand charge transfer). **PPy-DPP** shows absorption that is typical for a donor-acceptor conjugated polymer. The longer wavelength absorption at ~650 nm is due to the π-π* transition, and the absorption near 720 nm indicates interaction between the donor unit and acceptor unit along the main chain. **Ir-P1** shows an absorption spectrum that is very similar to that of **PPy-DPP** but with a blueshift of 50 nm. This is because the electron-withdrawing ability of the phenylpyridine unit is enhanced after the covalent coupling of iridium. **Ir1** shows short and weak absorption ($\varepsilon_{437} = 1.04 \times 10^4\,M^{-1}cm^{-1}$), but the two conjugated polymers exhibit strong absorption in the deep-red region (for **PPy-DPP**, $\varepsilon_{719} = 16.21 \times 10^4\,M^{-1}cm^{-1}$; for **Ir-P1**, $\varepsilon_{669} = 11.93 \times 10^4\,M^{-1}cm^{-1}$). **Ir2** and **DPP**'s absorption is no more than 550 nm, and the organic repeating unit, **QZL-DPP**, shows a maximal absorption at 610 nm. However, **Ir-P2**'s absorption is surprisingly red-shifted to 700 nm (maximal absorption at 660 nm, $\varepsilon_{660} = 9.00 \times 10^4\,M^{-1}cm^{-1}$), indicating iridium atom may be fully included in the backbone conjugation, consequently, forming a narrow-bandgap metallopolymer. Therefore, the problem of weak and short-wavelength absorption of Ir(III) compounds can be addressed by narrow-bandgap metallopolymerization strategy when the iridium atoms are covalently coupled or directly included in the conjugated polymer backbones, via very simple cross-coupling reactions. Their metallopolymeric nanoparticles were fabricated with minor modifications according to the reported work[50,51]. **Ir-P1** exhibits fluorescence emission close to 700 nm in tetrahydrofuran while **MPdot1** shows very weak and redshifted luminescence, as shown in

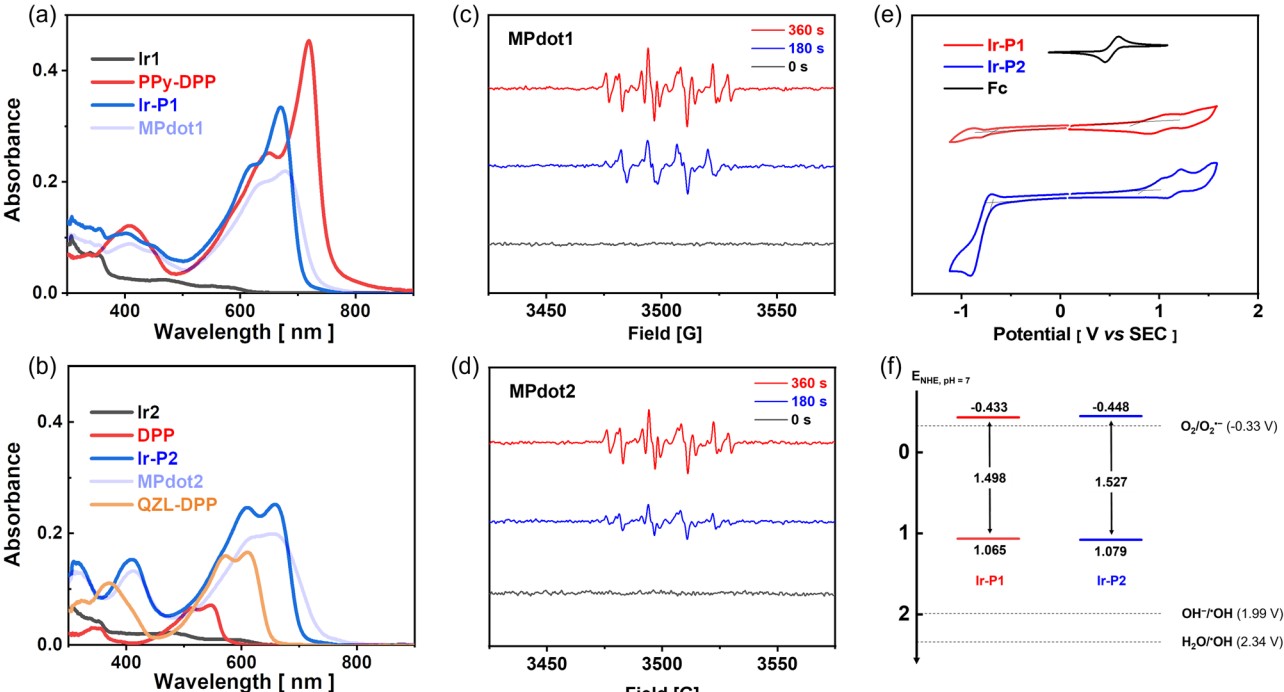

**Fig. 2 | Photophysical and photochemical properties of Ir(III) monomers and metallopolymers.** The absorption spectra of photosensitizers in (**a**, **b**). Experimental condition: for Uv/Vis spectra in dichloromethane, [**Ir1**] = [**PPy-DPP**] = [**Ir-P1**] = [**Ir2**] = [**DPP**] = [**Ir-P2**] = [**QZL-DPP**] = 2.8 μM, and in water, [**MPdot1**] = [**MPdot2**] = 2.8 μM with polymer calculated in repeating unit. The EPR (electron paramagnetic resonance) spectra data was obtained from an aqueous solution (DMSO : H₂O = 1 : 9, v/v) containing BMPO (5-t-butoxycarbonyl-5-methyl-1-pyrroline N-oxide) in (**c**, **d**). Experimental conditions: [**MPdot1**] = [**MPdot2**] = 0.5 mM, [BMPO] = 100 mM; irradiation: 680 nm laser (800 mW·cm⁻²). **e** Cyclic voltammogram of **Ir-P1** (red) and **Ir-P2** (blue). Experimental conditions: Scan rate 100 mV·s⁻¹ in anhydrous dichloromethane, [**Ir-P1**] = [**Ir-P2**] = 2 mg·mL⁻¹, [Bu₄NPF₆] = 0.2 M, saturated calomel electrode as a reference electrode, glassy-carbon electrode as a working electrode, and Pt wire as a counter electrode; Fc/Fc⁺ was used as an external reference. **f** Redox potential diagram of **Ir-P1** (red) and **Ir-P2** (blue).

Supplementary Fig. 3a, due to aggregation-caused quenching. **Ir-P2** emits weak fluorescence in solution and MPdot state as shown in Supplementary Fig. 3b. MPdots are relatively uniform with small sizes of approximately 20‑30 nm, as revealed by the dynamic light scattering (DLS) results and transmission electron microscopy (TEM) shown in Supplementary Fig. 3c, d. The energy-dispersive X-ray spectroscopy (EDS) of **MPdot1** and **MPdot2** provide strong evidence of iridium content in the nanoparticles as shown in Supplementary Fig. 3e, f. The surface is negatively charged with a potential of approximately −30.0 mV, which ensures good colloidal stability.

The qualitative analysis of the types of ROS generated by photosensitizers was based on electron paramagnetic resonance (EPR) spectral measurements. In the presence of the trapping reagent BMPO (5-t-butoxycarbonyl-5-methyl-1-pyrroline N-oxide) for $O_2^{\cdot-}$ and $\cdot OH$, and TEMP (3,3,5,5-tetramethyl-1-pyrroline N-oxide) for $^1O_2$, the observation of different signals provide evidence for the ROS types[52,53]. Both **MPdot1** and **MPdot2** generate a mixed $\cdot OH$ and $O_2^{\cdot-}$ signal in Fig. 2c, d. The $^1O_2$ signal can also be observed but with a much lower intensity for **MPdot1** and **MPdot2**, as shown in Supplementary Fig. 4a, b. The redox potential levels of two metallopolymers were measured to provide further support for the possibility of $O_2^{\cdot-}$ generation. The cyclic voltammetry indicated that they are thermodynamically feasible to generate $O_2^{\cdot-}$ as shown in Fig. 2e, f.

The facile tuning of absorption property is an important advantage of the metallopolymer strategy to develop a kind of candidate photosensitizers for practical application. Noticing that metallopolymers can inherit Type-I ROS generation, we further tested the generality of our strategy, by synthesizing another two metallopolymers, with different co-polymerization components, in which **Ir1** and blue/green-fluorescence emitting polymers (**PPy-F8** and **PPy-TBT**) were combined as complementary to red-fluorescence emitting polymer (**PPy-DPP**). Surprisingly, the metallopolymers' MPdot also exhibits a clear Type-I ROS signal in the EPR test as shown in Fig. 3b, c and they can also generate ROS times over Ce6 in the presence of 473 nm laser. Based on current results, the strategy works well in blue, green, and deep-red regions, and can further extend absorption to NIR regions. We consider tuning the ratio of **Ir1** and **DPP** units to manipulate the absorption behavior of the resulting metallopolymers. **Ir-P44** and **Ir-P37**, in which the iridium units take up 44.4% and 37.5% of total structure units, respectively, exhibit increased absorption in NIR regions, compared with **Ir-P1** where the iridium units constitute 50% of the total units. As shown in Fig. 3d, e, clear Type-I ROS signals in EPR could be observed. DPBF (1,3-diphenylisobenzofuran) is widely used as an indicator for ROS detection with a high reaction rate. DPBF can react with other radicals and oxidants, and, thus, can be used to detect the total ROS generated. We conducted **Ir-P44** and **Ir-P37**'s ROS

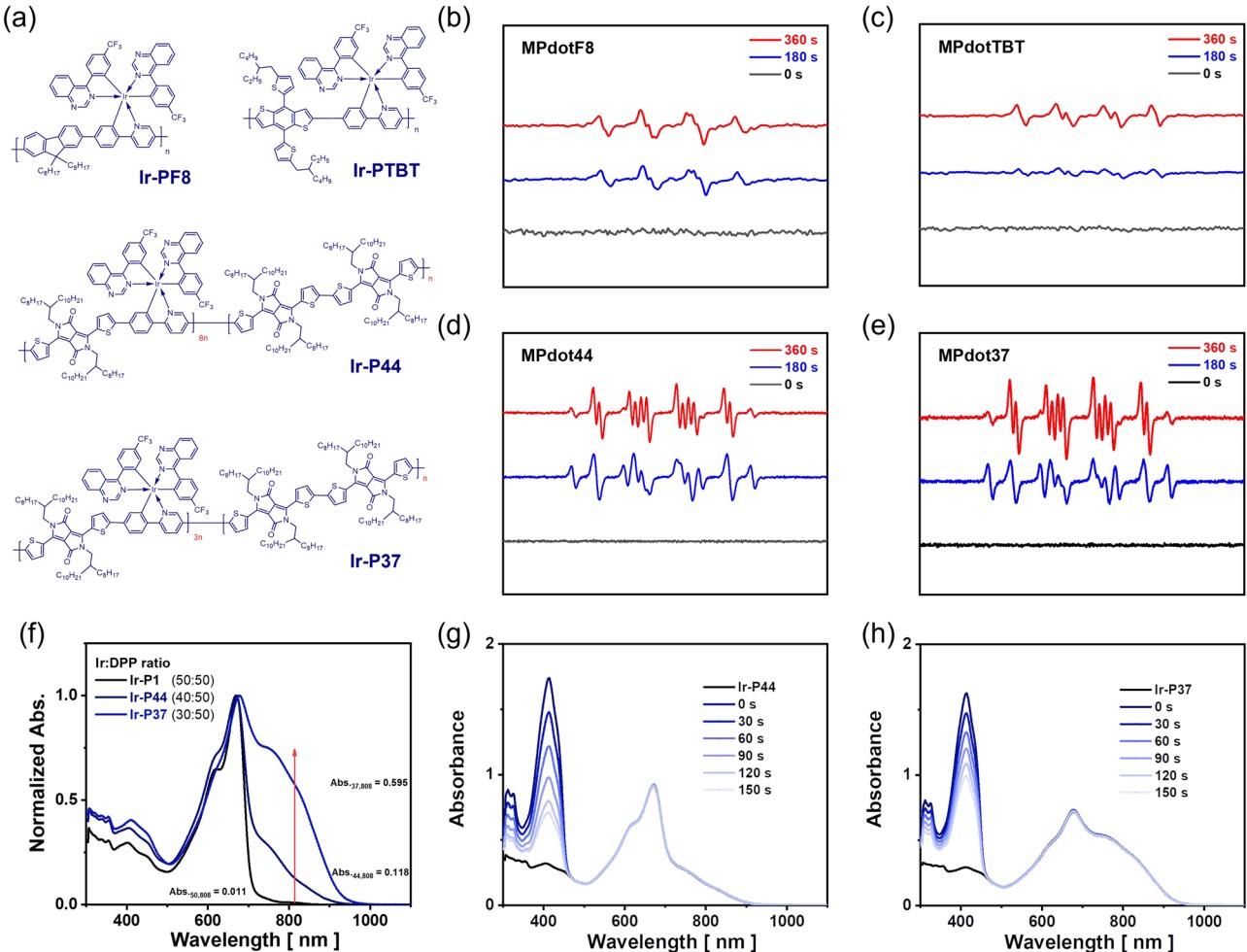

**Fig. 3 | More metallopolymers with Type-I photochemical pathways. a** The structure of **Ir-PF8, Ir-PTBT, Ir-P44** and **Ir-P37**. The EPR data was obtained from an aqueous solution (DMSO : H₂O = 1 : 9, v/v) containing BMPO. Experimental conditions: [**MPdotF8**] = [**MPdotTBT**] = [**MPdot44**] = [**MPdot37**] = 0.5 mM, [BMPO] = 100 mM; irradiation: white LED light (50 mW·cm⁻²) for (**b**) **MPdotF8** and (**c**) **MPdotTBT** and 808 nm laser (1 W·cm⁻²) for (**d**) **MPdot44** and (**e**) **MPdot37**. **f** The normalized absorption of **Ir-P1, Ir-P44,** and **Ir-P37** in dichloromethane. The absorption spectra of DPBF in dichloromethane solutions. **g Ir-P44**; **h Ir-P37**. Experimental conditions: [DPBF] = 50 µM, [**Ir-P44**] = [**Ir-P37**] = 14.0 µM with 808 nm laser (1 W·cm⁻²) and recorded every 30 s.

generation experiment with DPBF, as shown in Fig. 3g, h and the Type-I ROS generation was further confirmed in vitro with dihydroethidium (DHE) as an $O_2^{\cdot-}$ probe (Supplementary Figs. 5 and 6). These metallopolymers show reasonable ROS generation efficiency under 808 nm irradiation. However, we observed a strong photothermal effect of NIR metallopolymer dots, as shown in Supplementary Fig. 7, which is a very common property for NIR materials. To avoid the strong interference of the photothermal effect on tumor elimination, we selected **MPdot 1** and **MPdot 2** with absorption around 670 nm (in a biological window) for in vitro and in vivo study.

The quantitative ROS generation abilities of these monomeric and metallopolymeric photosensitizers were evaluated with Ce6 as a reference in the presence of laser irradiation. In this testing condition, **Ir1** exhibits an undetected decrease at 411 nm, which is expected and ascribed to the absorption region being not long enough to reach 680 nm, as shown in Fig. 4a. **PPy-DPP** only shows a very weak ROS generation in Fig. 4b. As shown in Fig. 4c, over an interval of 30 s of irradiation, DPBF's absorption at 411 nm decreases rapidly, indicating the generation of ROS in the system. **Ir-P1** exhibits efficient ROS generation, which is over 80 times higher than that of its mother polymer **PPy-DPP** without iridium center. As shown in Fig. 4d, e, the monomer **Ir2** and **DPP** neither fail to generate ROS. However, **Ir-P2** consumes DPBF probe at a high rate in the presence of laser irradiation. The

physical doping or mixing system of several photosensitizers were investigated to check if there would be some sort of synergistic effects to improve the ROS generation process. Disappointingly, as shown in Fig. 4g, h, no obvious absorption changes around 411 nm can be detected, compared to their individual ROS generation.

The monomeric and metallopolymeric photosensitizers are also evaluated in the presence of a 473 nm laser with Ce6 as a reference. All results are listed in Supplementary Table 2. All these facts prove the usefulness and necessity of the narrow-bandgap metallopolymerization strategy to boost high ROS generation. First, coupling or including iridium atoms to polymer backbones to form metallopolymers can significantly improve ROS generation. For **PPy-DPP**, even with the strongest molar absorption coefficient of all the photosensitizers, the lack of a catalytic center fails to generate ROS efficiently after harvesting light. Physical mixing of **PPy-DPP** and **Ir1** fails to improve ROS generation. As a result, chemical bonds that lock iridium atoms in the conjugation are very important. In the **Ir-P2** case, the iridium atom becomes more important for being the linkage of organic segments. The incorporation of an iridium atom makes the extension of the absorption to deep-red regions possible and undertakes the catalysis since **QZL-DPP** hardly generates ROS in the presence of a 680 nm laser as shown in Supplementary Fig. 8a. In addition, the high molar absorption coefficient of metallopolymer favors high ROS generation.

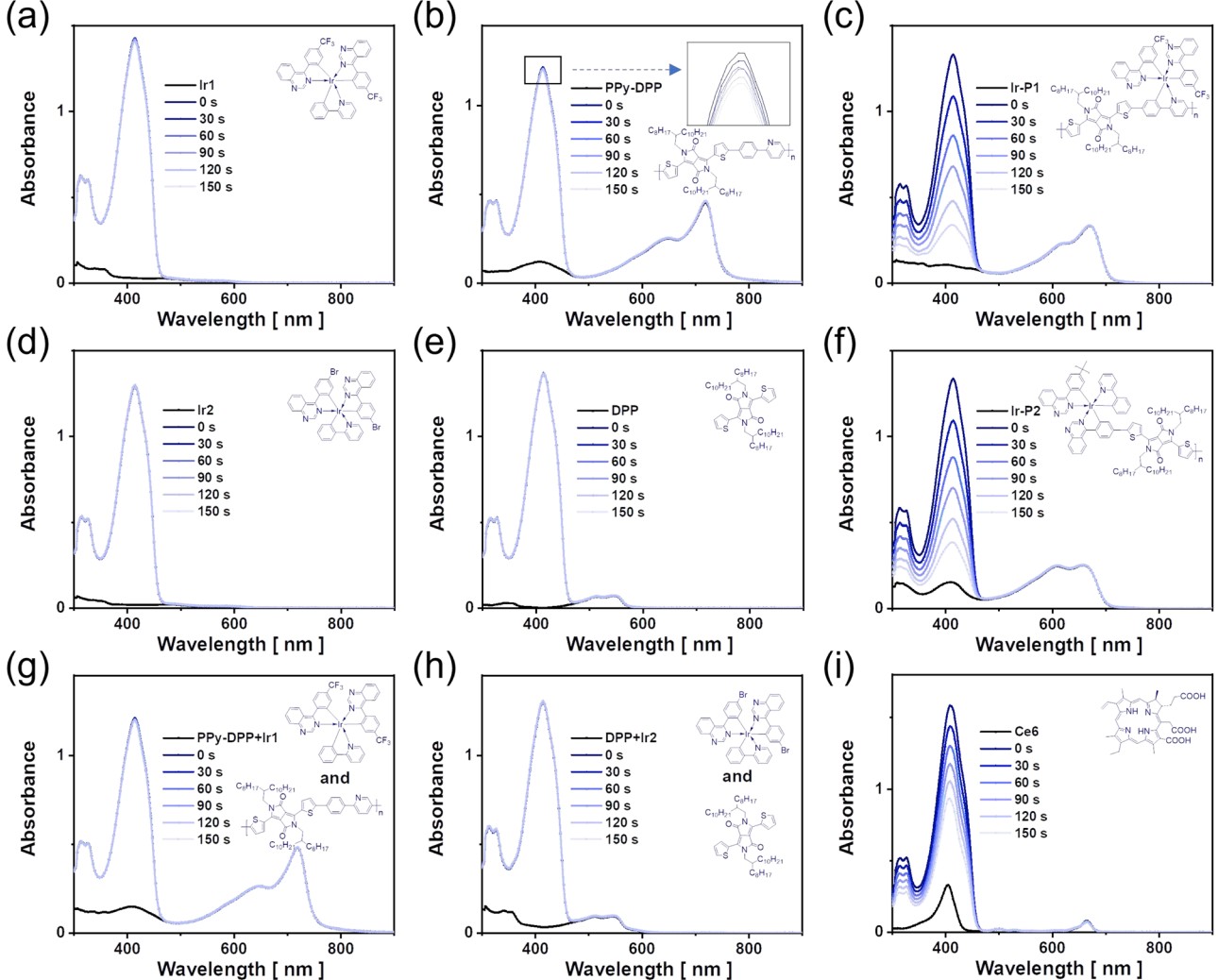

**Fig. 4 | ROS efficiency evaluation in solution.** The absorption spectra of DPBF in dichloromethane solutions. Experimental conditions: [DPBF] = 50 µM, [**Ir1**] = [**PPy-DPP**] = [**Ir-P1**] =[**Ir2**] = [**DPP**] = [**Ir-P2**] = [Ce6] = 2.8 µM with a 680 nm laser (5 mW·cm⁻²) and recorded every 30 s. **a** Ir1; **b** PPy-DPP; **c** Ir-P1; **d** Ir2; **e** DPP; **f** Ir-P2; **g** the blend solution of **Ir1** and **PPy-DPP**; **h** the blend solution of **Ir2** and **DPP**; **i** Ce6.

Even in the presence of a 473 nm laser, which all photosensitizers can absorb, two metallopolymers exhibit the highest ROS generation behavior, as shown in Supplementary Fig. 8b–h. High absorption in all light regions is responsible for these results. High absorption at 680 nm would also be the main explanation that two metallopolymers generate a higher absolute amount of ROS than that of Ce6, even though the ROS quantum yields of **Ir-P1** (33%) and **Ir-P2** (37%) are lower than that of Ce6. In addition, their MPdot exhibited higher photostability than Ce6 for a long period of exposure to stronger irradiation as shown in Supplementary Fig. 9, which can bring possible convenience in terms of less frequent drug dosing and a prolonged curing period. In short, narrow-bandgap metallopolymers have more expanded conjugation and a longer persistence length of polymer chain[42,43], and thus greatly enhance the photo-absorption properties and finally contribute to rapid ROS generation. We also conducted transient absorption experiments and found that the introduction of Ir(III) unit to conjugated polymer dramatically prolongs the charge lifetimes, which also favors photo conversion in the ROS generation, as shown in Supplementary Fig. 10.

To further determine the ROS type for **Ir-P1** and **Ir-P2** in solution, several scavengers were used for ROS detection in a THF/water medium. To be noted, DPBF shows very poor solubility in water and Tiron (disodium 4,5-dihydroxy-1,3-benzenedisulfonate) poor in organic solvents, so a blend solution system is used and two metallopolymers' nanoparticles, **MPdot1** and **MPdot2** are used consequently. Taking **MPdot1** as an example, as shown in Fig. 5a, without any scavenger, the MPdot consumes the probe in the presence of a 680 nm laser. However, upon the addition of Tiron, the scavenger for $O_2^{\cdot-}$[54], the decrease in absorption at 411 nm is suppressed, indicating that the consumption of DPBF is inhibited, as shown in Fig. 5b. To ensure that Tiron does not scavenge $^1O_2$, Ce6 was used as the $^1O_2$ generator, and in the presence of Tiron, almost no difference is observed, as shown in Supplementary Fig. 11a, b. However, the ROS generation remains unchanged upon the addition of *t*-butanol, the scavenger for $\cdot$OH as shown in Fig. 5c. The

time-dependent normalized absorption at 411 nm is summarized in Fig. 5d. These results lead to the conclusion that the ROS generated by **MPdot1** are mainly composed of $O_2^{\cdot-}$, part of the $^1O_2$, and almost no $\cdot$OH. The $\cdot$OH signal in EPR spectra can be accredited to the transformation of $O_2^{\cdot-}$ to $\cdot$OH[55] or BMPO-OOH to BMPO-OH since $\cdot$OH generation is not thermodynamically possible based on the redox potential level. The fluorescence spectra also provide evidence for ROS type determination. Dihydrorhodamine 123 (DHR 123) is another fluorescence probe that can be used for the detection of $O_2^{\cdot-}$. As shown in Fig. 5e, upon laser irradiation, the fluorescence at 525 nm increases rapidly, indicating the detection of $O_2^{\cdot-}$ in the testing solution. With increasing irradiation time, the fluorescence intensity is enhanced 65 times higher than the initial intensity; however, after irradiating a solution containing only DHR 123, the autofluorescence of the probe is observed with nearly no change, as shown in Supplementary Fig. 11c. When Singlet Oxygen Sensor Green (SOSG) is used as a fluorescent probe, the fluorescence intensity at 525 nm remains with minor changes as shown in Fig. 5f. Since SOSG is highly specific to $^1O_2$, it is very likely that only a small portion of $^1O_2$ is generated. The same tests have been conducted using **MPdot2** as photosensitizers with other testing parameters unchanged, and similar phenomena were observed, as shown in Supplementary Fig. 12. To further confirm the $O_2^{\cdot-}$ generation, 2-(4-iodophenyl)−3-(4-nitrophenyl)-5-(2,4-disulfophenyl)-2H- tetrazolium sodium salt (WST−1) was used as the spectrophotometric probe. Upon the irradiation of 680 nm laser, WST−1 slowly evolved and absorption at 435 nm increased, indicating the generation of $O_2^{\cdot-}$ as shown in Supplementary Fig. 13. In addition, we also confirmed the total ROS generation in solution with freshly prepared DCFH, by hydrolyzing H₂DCFDA with 0.01 M NaOH solution. In the presence of a 680 nm laser, the green fluorescence of DCF was observed upon activating **MPdot1** or **MPdot2**, as shown in Supplementary Fig. 14a, b. However, when Hydroxyl Radical and Peroxynitrite Sensor (HPF) was used, no obvious fluorescence intensity was observed, as shown in Supplementary Fig. 14d, e. Therefore, the ROS generated by **MPdot1**

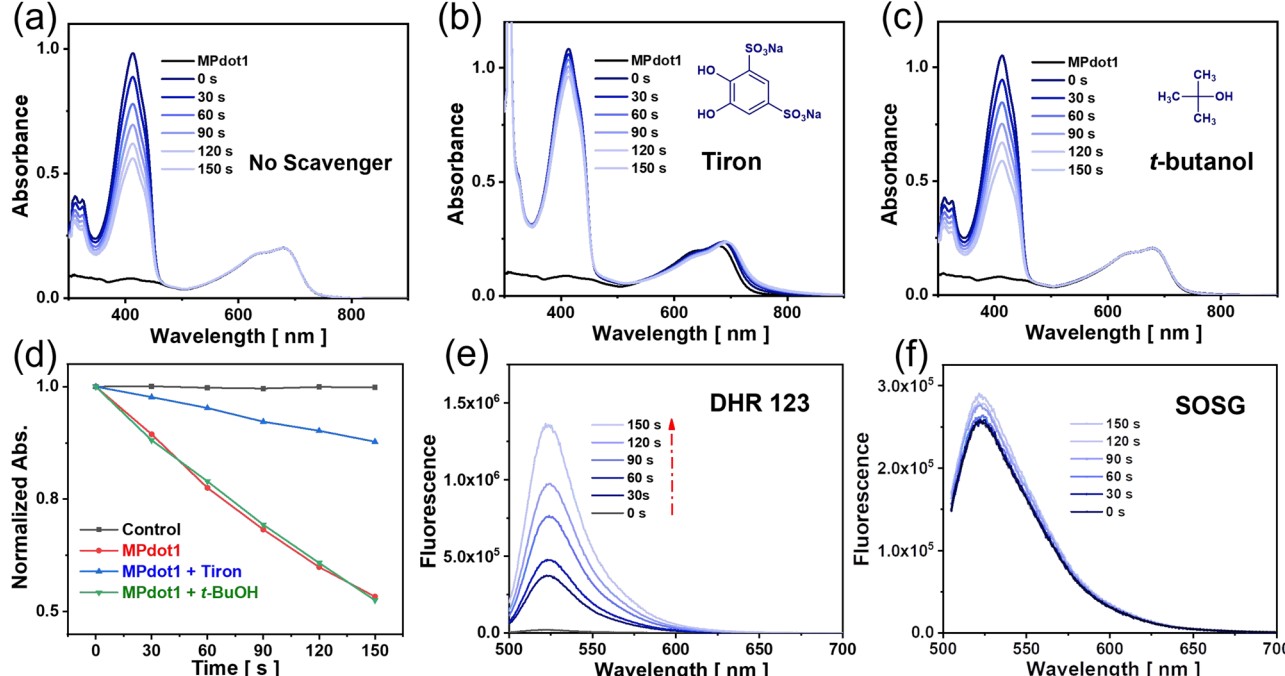

**Fig. 5 | ROS type detection in solution.** Absorption spectra for DPBF in tetrahydrofuran/water (1 : 1, v/v) solution. Experimental conditions: [DPBF] = 50 μM, [**MPdot1**] = 2.8 μM, [Tiron] = [*t*-butanol] = 50 mM with a 680 nm laser (20 mW·cm⁻²) and recorded every 30 s. **a** No scavenger; **b** Tiron (disodium 4,5-dihydroxy-1,3-benzenedisulfonate); **c** *t*-butanol; **d** The time-dependent normalized absorption

was measured at 411 nm in the control group, MPdot group, and MPdot group with scavengers. The fluorescence spectra for ROS probes in water. Experimental conditions: [DHR 123] = [SOSG] = 20 μM, [**MPdot1**] = 2.8 μM with a 680 nm laser (20 mW·cm⁻²). **e** DHR 123 (Dihydrorhodamine 123); **f** SOSG (Singlet Oxygen Sensor Green).

and **MPdot2** upon light irradiation is proven to mainly be $O_2^{\cdot-}$ with a minor portion of Type-II ROS.

## In vitro study

The therapeutic effect of **MPdot** was validated at the cellular level. For better tracking of **MPdot1**, another blue fluorescence emitting polymer, **PPy-F8**, was doped into **MPdot1**. To ensure the uptake of **MPdot1**, it was further coated with the cell-penetrating peptide R8 via static electric interaction to afford positively charged **MPdot1r**. The post-modification of **MPdot1r** has minor effects on the particle size, but the surface charge can be tuned with different amounts of the cell-penetrating peptide with data summarized in Supplementary Fig. 15. Cells exhibit stronger blue fluorescence after incubation with R8-decorated MPdot, compared to that of the group treated with pristine MPdot, indicating that post-decoration promotes the uptake of nanoparticles.

The ROS generations by **MPdot1r** and **MPdot2r** without mixing with blue-emitting polymer were evaluated in HeLa cells consequently. Taking **MPdot1r** as an example, as shown in Fig. 6a$_1$, a$_3$, bright green fluorescence can be detected in cells after laser irradiation using H$_2$DCFDA as the probe with optimized experimental conditions, indicating total ROS generated in cells. The same fluorescence increase was observed in flow cytometry, as shown in Fig. 6a$_4$. Dihydroethidium (DHE) is a highly specific $O_2^{\cdot-}$ probe whose product can bind with DNA and emit red fluorescence. Both in confocal imaging and flow cytometry, strong red fluorescence was detected, showing the generation of $O_2^{\cdot-}$. Similar results have been observed using DHR 123 as the $O_2^{\cdot-}$ probe. It has been confirmed that sole MPdot or laser irradiation failed to increase fluorescence intensity as shown in Supplementary Figs. 16−18. In addition, to further confirm the ability of $O_2^{\cdot-}$ generated intracellularly following 680 nm laser irradiation, an $O_2^{\cdot-}$ scavenger (1,4-benzoquinone, BQ) was used and successfully inhibited the turn-on fluorescence of DHR 123 as shown in Supplementary Fig. 19. **MPdot2r** also shows quite similar ROS generation behavior in cells, as summarized in Supplementary Figs. 20−24. These results indicate that

**MPdot1r** and **MPdot2r** can rapidly generate a large amount of $O_2^{\cdot-}$ under laser stimulation in cells.

Encouraged by the strong $O_2^{\cdot-}$ generating ability, we applied the MPdots for PDT and tested the antitumor effects in vitro. First, apoptosis in different groups was evaluated by flow cytometry. As shown in Fig. 7a−d, compared with the Laser+/**MPdot1r**- group, and others, the Laser+/**MPdot1r**+ group exhibits obvious late apoptosis. To better understand the mechanism for the therapeutic effect of **MPdot1r**, the expression of BcL-2, an important indicator protein for apoptosis, was monitored in different groups by western blot assay (Fig. 7e and full scan in Supplementary Fig. 25). The results show that the expression of BcL-2 protein in the Laser+/**MPdot1r**+ group is significantly decreased, indicating that apoptosis occurs in the Laser+/**MPdot1r**+ group. To more intuitively show the anti-tumor effect of MPdot, we stained all cells with the Calcein-AM/PI staining kit after the treatments and collected fluorescent images of the cells by confocal microscopy. As shown in Fig. 7f$_{1-12}$, with increasing **MPdot1r** concentration, the proportion of red fluorescence in the vision also ascends, indicating that the anti-tumor effect of MPdot-related PDT is stronger. To further illustrate the outstanding anti-tumor effect of MPdot, we detected the viability of HeLa cells treated with different concentrations of MPdot after laser stimulation under normoxic and hypoxic conditions, as shown in Fig. 7g, h. At a normal $O_2$ level (21%), **MPdot1r** exhibits a low IC$_{50}$ of 7.00 μg/mL. Notably, we found that **MPdot1r** still shows an excellent antitumor effect with an IC$_{50}$ of 7.16 μg/mL after laser irradiation, even in a hypoxic environment where the $O_2$ level is as low as 1%. 4T1 cell lines can also be efficiently killed with the treatment of **MPdot1r**, as shown in Supplementary Fig. 26. **MPdot2r** exhibits similar light toxicity as **MPdot1r** in the presence of a 680 nm laser for HeLa and 4T1 cell lines in normoxic and hypoxic environments, as shown in Supplementary Fig. 27. On the contrary, **Ir1, Pdotr** of **PPy-DPP, Ir2**, and **QZL-DPP** show almost no light toxicity under 680 nm laser irradiation due to no absorption of this light or poor ROS generation, which are presented in Supplementary Fig. 28. These results consist with former ROS detection in solution.

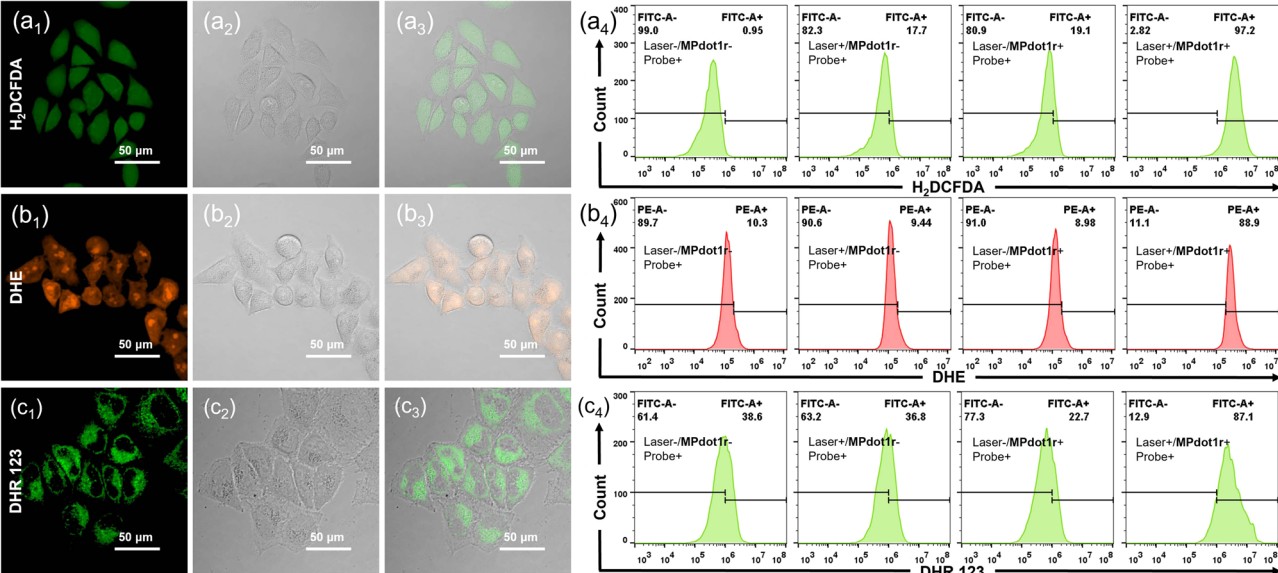

**Fig. 6 | ROS assay in vitro with probes.** Total ROS detection with H$_2$DCFDA (**a$_1$**–**a$_4$**). Experimental conditions: [**MPdot1r**] = 60 μg/mL with preincubation for 20 h, [H$_2$DCFDA] = 5 μM with incubation for 60 min, 680 nm laser irradiation (40 mW·cm$^{-2}$) for 10 min. $O_2^{\cdot-}$ detection with DHE (**b$_1$**–**b$_4$**). Experimental conditions: [**MPdot1r**] = 20 μg/mL with preincubation for 20 h, [DHE] = 5 μM with incubation for 30 min, 680 nm laser irradiation (40 mW·cm$^{-2}$) for 5 min. $O_2^{\cdot-}$ detection with DHR 123 (**c$_1$**–**c$_4$**). Experimental conditions: [**MPdot1r**] = 20 μg/mL with preincubation for 20 h, [DHR 123] = 5 μM with incubation for 30 min, 680 nm laser irradiation (40 mW·cm$^{-2}$) for 5 min. (**a$_1$, b$_1$**, and **c$_1$**). The FICT channel; (**a$_2$, b$_2$**, and **c$_2$**). The transmitted light differential interference (TD) channel; (**a$_3$, b$_3$**, and **c$_3$**). The merged channel. (**a$_4$, b$_4$**, and **c$_4$**). Flow cytometry for ROS generation. All experiments are repeated three times independently with similar results.

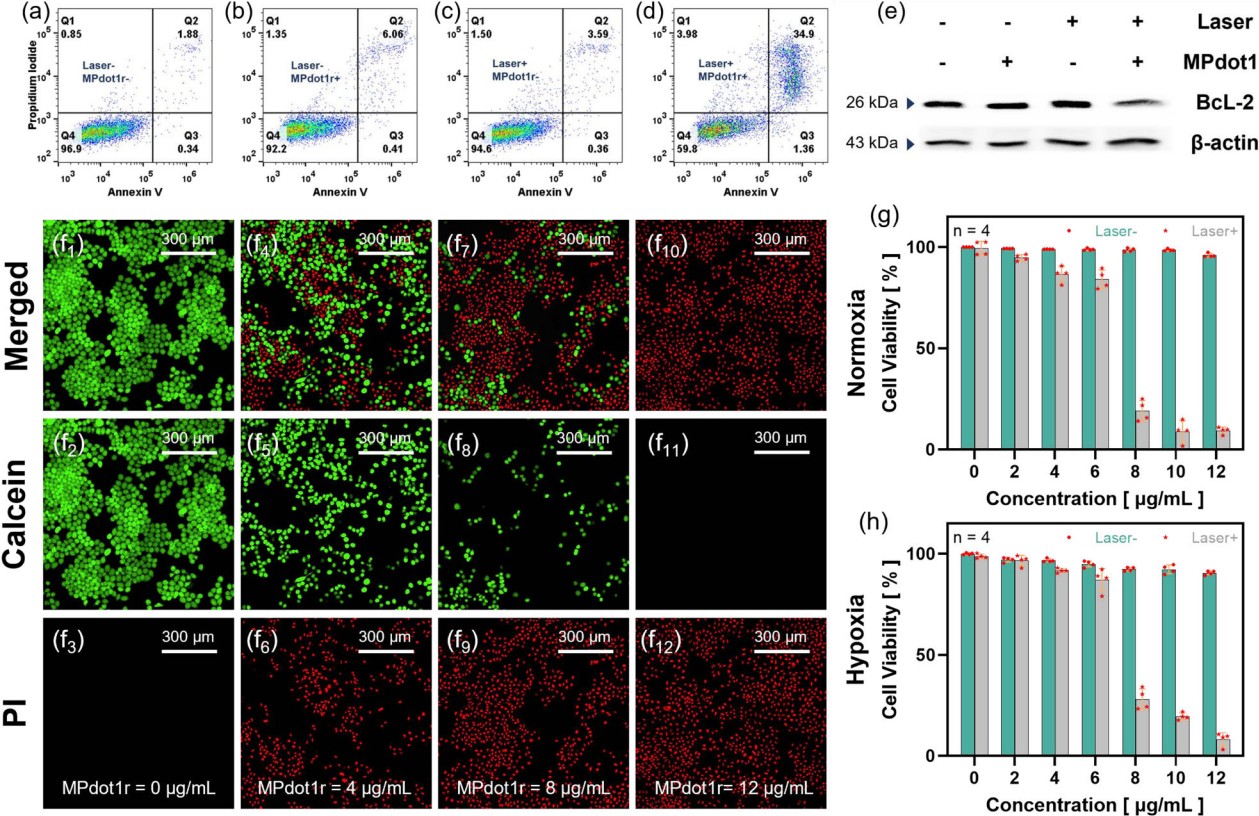

**Fig. 7 | Tumor cell killing evaluation in vitro.** Apoptosis assay using flow cytometry after staining with an Annexin V/propidium iodide (PI) kit. **a** The laser-/**MPdot1r**- group; **b** The laser-/**MPdot1r**+ group; **c** The laser + /**MPdot1r**- group; **d** The laser+/**MPdot1r**+ group. **e** Western blot analysis of the BcL-2 protein using β-actin as an internal reference, repeated three times independently with similar results. Living/dead cells were stained by Calcein-AM/PI. (**f₁, f₂, f₃**). [**MPdot1r**] =

0 µg/mL, (**f₄, f₅, f₆**). [**MPdot1r**] = 4 µg/mL, (**f₇, f₈, f₉**). [**MPdot1r**] = 8 µg/mL, (**f₁₀, f₁₁, f₁₂**). [**MPdot1r**] = 12 µg/mL. Cell viability assay with varying concentrations of MPdot. **g** Normoxia ($V\%_{O_2}$ = 21%) and **h** hypoxia ($V\%_{O_2}$ = 1%), respectively. Data are presented as mean ± SD derived from $n$ = 4 independent samples. All experiments are repeated three times independently with similar results.

## In vivo photodynamic therapy

Next, we examined the antitumor effect and bio-safety of **MPdot1** using the 4T1 breast tumor model. Nanoparticles can passively enrich tumors due to enhanced permeability and retention effects. However, recent studies showed that only a small amount of the administered nano-formulated drug dose ends up in solid tumors[56,57]. Currently, many researchers are trying to explore good methods to improve nanoparticle enrichment in tumors. To enhance the targeting ability of MPdot in vivo, the **MPdot1** is modified with c(RGDyK) peptide through bioconjugation with -COOH group from PSMA as illustrated in Fig. 8a and Supplementary Fig. 29 to form **MPdot1c**. We studied the enrichment of nanoparticles in tumors using a photoacoustic (PA) technique. Firstly, the PA behavior of **MPdot1** and **MPdot2** in solution was analyzed and the results were shown in Supplementary Fig. 30. Photoacoustic image of the tumor site was taken 0, 3, 7, 12, and 24 h after the mice were intravenously injected with **MPdot1c** via the tail vein (Fig. 8c). Increased PA signal after 7 h indicated tumor accumulation of **MPdot1c**, which reached the highest at 12 h post-injection. c(RGDyK) peptide-modified **MPdot1c** with a PA intensity of 0.636 at 12 h exhibited better accumulation in the tumor than that of unmodified **MPdot1** (0.328 at 12 h) and **MPdot2c** also showed enhanced accumulation in the tumor (Supplementary Fig. 31). As a result, 12 h after the injection of **MPdot1c**, mice in laser groups were irradiated with 680 nm laser at 300 mW·cm⁻² for 10 min. After two cycles of treatment, mice were further observed for 5 days with tumor volume monitored (Fig. 8d and Supplementary Fig. 32) and sacrificed with organs and tumors

harvested on the 17th day. Compared with the growing tumor volume in other groups, tumors in the Laser+/**MPdot1c**+ group were inhibited, especially tumors that disappeared in four mice, with a tumor weight ratio of 24.84 (weights in Laser-/**MPdot1c**+ groups *vs* Laser+/**MPdot1c**+), as summarized in Fig. 8g. In the same experimental conditions, the in vivo therapeutic effects of **MPdot2c** were evaluated as well, tumors can be efficiently inhibited after treatment (tumor weight ratio, 12.04), as shown in Supplementary Fig. 33. These outstanding therapeutic results may be highly related to the hypoxia-tolerant behavior of Type-I photosensitizers[58,59], since the tumors are normally featured as hypoxic. We re-confirmed this phenomenon with immunofluorescence staining and photoacoustic imaging techniques (Supplementary Fig. 34). The hypoxic tumor microenvironment is unfavorable to the production of ROS by the Type-II photosensitizer. To validate this hypothesis, one Type-II polymeric photosensitizer (**p-BODIPY-F**) with similar absorption behavior to **Ir-P1** was tested. A similar ROS generation rate of the **p-BODIPY-F**'s nanoparticle, **PBdot**, in solution, was realized, compared to **MPdot1**, with increased concentration and laser power, and the scavenging test helped to confirm its Type-II ROS generation, as shown in Supplementary Fig. 35a–d. However, in vitro experiments showed that the cell killing effects relaid heavily on O₂ concentration, and this O₂-dependence of Type-II **PBdotc** (**PBdot** with c(RGDyK) modification) may be responsible for poor in vivo therapeutic effects as shown in Supplementary Fig. 35f–h, and the calculated tumor weight ratio was 1.25, which was much lower than that of **MPdot1c** and **MPdot2c**, as shown in Fig. 8g.

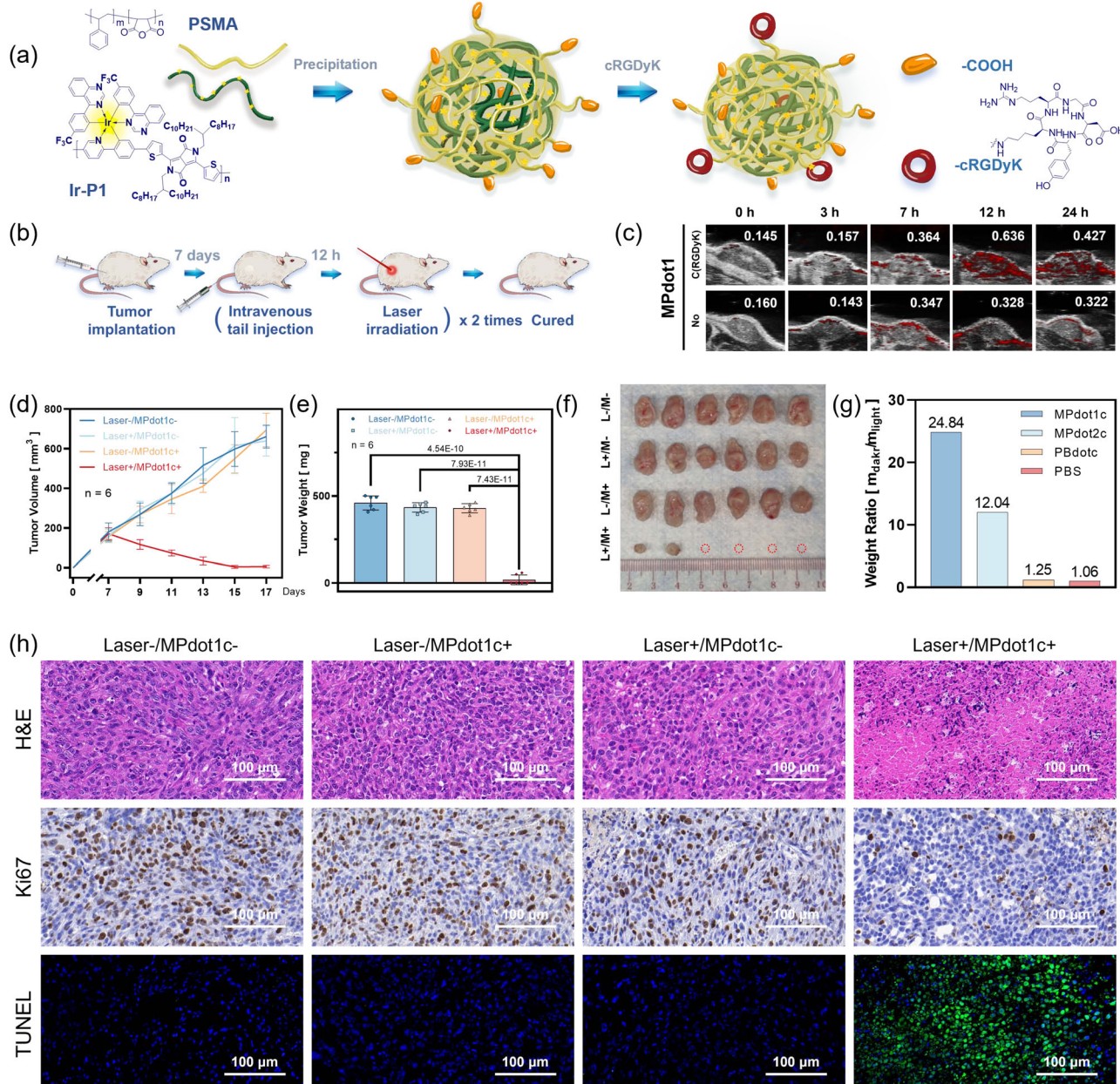

**Fig. 8 | Tumor cell killing evaluation in vivo. a** Illustration of the process for c(RGDyK)-modified **MPdot1c**. **b** Illustration of the process for therapy. **c** Photoacoustic image at tumor site 0, 3, 7, 12, 24 hours after tail vein intravenous injection with **MPdot1c** or **MPdot1** (100 μL, 400 μg/mL, in PBS). **d** Tumor volume was measured at the indicated day with a caliper and calculated using the following equation: $V = \frac{\pi}{6} \times A \times B^2$, $A$ is long diameter and $B$ is short diameter of the tumor. **e** Tumor weight measured after dissection. Data are presented as mean ± SD from

$n = 6$ mice. Numbers of pairwise comparison: $P$ value of one-sided $t$ test and no adjustments are made for multiple comparisons. **f** Tumor photo for mice in each group. **g** The ratio of tumor weight (tumors treated with **MPdot** in dark groups $vs$ tumors in laser groups). **h** The hematoxylin and eosin staining (H&E) staining, Ki67 immunohistochemistry, and the terminal deoxynucleotidyl transferase dUTP nick-end labeling (TUNEL) staining of tumor sections following different treatments, $n = 3$ tissue slices were prepared and similar results were observed.

The hematoxylin and eosin staining (H&E staining) of tumors further demonstrated the anti-tumor efficacy of **MPdot1c**-related PDT. Immunohistochemical staining for Ki67 revealed decreased cell proliferation and terminal deoxynucleotidyl transferase dUTP nick-end labeling (TUNEL) staining showed increased apoptosis following **MPdot1c** or **MPdot2c** and laser treatment in Fig. 8h and Supplementary Fig. 36. Moreover, the H&E staining of major organs (heart, liver, spleen, lung, and kidney) supported the bio-safety of **MPdot1c** and **MPdot2c** as shown in Supplementary Fig. 37. These results prove that MPdots are excellent photosensitizers with outstanding anti-tumor effects in hypoxic tumor microenvironments and good bio-compatibility.

In summary, we have demonstrated an effective metallopolymer strategy for exploring high-performance Type-I photosensitizers with strong absorption in the deep-red region. Unlike monomeric Ir(III) complexes, metallopolymers **Ir-P1** and **Ir-P2** show strong absorption that extends to the deep-red region. Compared with the mother polymer **PPy-DPP** without iridium catalytic center, the ROS generation efficiency of **Ir-P1** is significantly improved (over 80 times). The metallopolymer nanoparticles can generate ROS intracellularly upon 680 nm laser irradiation, resulting in sufficient phototoxicity to induce cell apoptosis. Type-I ROS-generating nature facilitates PDT outcomes even under hypoxic conditions, which endows **MPdots'** excellent therapeutic effects in vivo. This strategy is highly effective regardless

of whether the iridium atoms are covalently coupled or directly included in the polymer backbone, and this strategy also works efficiently in blue, green, deep-red, and NIR regions to generate Type-I ROS, indicating its potential and generality in exploring metallopolymer-based photosensitizers. This strategy is anticipated to find broad use in exploring metallopolymers with Type-I photochemistry and intense absorbance in the NIR region by judiciously tailoring the molecular structures of iridium complexes and organic building blocks.

## Methods

HeLa cells and 4T1 cells were obtained from the Faculty of Health Sciences, University of Macau. Female BALB/c mice (5-week old) were provided by the animal facility of the University of Macau. All animal procedures were approved by the Institutional Animal Care and Use Committee of the University of Macau (approval number: UMARE-013-2022). Sex was considered in the study design since 4T1 is a mammary carcinoma originally derived from a spontaneously arising mammary tumor in BALB/cfC3H mice. The maximal tumor burden permitted by the UM Animal Ethics Committee is 1300 mm$^3$ and the maximal tumor size/burden in this study was not exceeded. The details of the synthesis are provided in the Supplementary Information.

### Gel permeation chromatography

A 100 µL aliquot of the polymer solution, which had a concentration of 1 mg/mL in tetrahydrofuran (THF), was processed through the Malvern Viscotek TDA 305 system at a flow rate of 1 mL/min. The data gathered was subsequently analyzed using the Malvern Viscotek software and calibrated with a standard curve. This standard curve was derived from the testing of standard polystyrenes with diverse molecular weights (170,000, 103,500, 77,000, 29,600, 13,400, 6040, 2560, and 1200 Da, respectively).

### Uv/Vis absorption and fluorescence spectra

The compounds **Ir1, Ir2, DPP**, and **QZL-DPP** were dissolved in dichloromethane (DCM) to form a 1 mM solution, which was subsequently diluted to a concentration of 2.80 µM. Similarly, **Ir-P1, PPy-DPP**, and **Ir-P2** were dissolved in DCM to make a 1 mg/mL solution. For **Ir-P1**, this metallopolymer solution was then diluted to achieve concentrations of 5.00 µg/mL (equivalent to 2.80 µM in one repeating unit), 2.92 µg/mL for **PPy-DPP**, and 4.54 µg/mL for **Ir-P2**. The UV/Vis spectra were obtained by measuring 2 mL of each of these solutions. For the fluorescence spectra, the solutions were additionally diluted by a factor of 5 before measurement.

### MPdot fabrication

MPdot was fabricated by the precipitation method. **Ir-P1, Ir-P2, Ir-P44, Ir-P37, Ir-PF8, Ir-PTBT, PPy-DPP, PPy-F8, PPy-TBT** and PSMA (Sigma-Aldrich, 442402-250G-A, #08728HN) were dissolved with THF at a concentration of 1 mg/mL, respectively, and stored at −20 °C for further use. 250 µL **Ir-P1** solution and 100 µL PSMA solution were added into a glass bottle and diluted with THF to 5 mL and injected into 25 mL of deionized water under ultrasonication and sonicated for 10 mins to afford a homogenous dark green solution. The solution was dialyzed with deionized water (2 L × 6 times) for 48 h with a cut-off of 100,000 Da. After dialysis, the solution was filtered with a 0.22 µm filter and concentrated under reduced pressure to afford a dark green solution. The concentration of the solution is based on the metallopolymer and calibrated with linear-fitted Uv/Vis concentration-dependent absorption spectra. MPdot solution can be stored at 4 °C for further use.

The blue fluorescence emitting MPdot was fabricated similarly, with a blend solution of 250 µL **Ir-P**: 50 µL **PPy-F8**: 100 µL PSMA.

**PPyF8**

Other polymeric nanoparticles were fabricated in a similar procedure.

### MPdot size and surface potential

MPdot solution was diluted to 50 µg/mL and 2 mL was measured on dynamic light scattering (Malven Zetasizer Nano ZS).

### Electron paramagnetic resonance spectra

The EPR measurements were conducted using a Bruker Model A300 at ambient temperature. All samples were prepared under identical conditions, with PSs (0.5 mM) and BMPO or TEMP (100 mM) in a total volume of 200 µL. Different light sources were employed for different compounds: a white light source (50 mW·cm$^{-2}$) for **Ir1, Ir2, MPdotF8**, and **MPdotTBT**; a 660 nm laser (800 mW·cm$^{-2}$) for **MPdot1** and **MPdot2**; and an 808 nm laser (1 W·cm$^{-2}$) for **MPdot44** and **MPdot37**. Before the EPR spectra measurements, the samples were exposed to light for 5 minutes. For time-dependent spectra, the samples were irradiated, and the spectra were recorded at each time interval. Control groups were prepared in the same manner but without any light exposure.

### Cyclic voltammetry

The cyclic voltammetry measurements were conducted on a HY 1550 A mini electrochemical analyzer. All samples (2 mg·mL$^{-1}$) were evaluated in anhydrous dichloromethane at a scan rate of 100 mV·s$^{-1}$. Bu$_4$NPF$_6$ was used as an electrolyte at a concentration of 0.2 M. Saturated calomel electrode was used as a reference electrode, with a glassy-carbon electrode used as a working electrode, and Pt wire as a counter electrode; and Fc/Fc$^+$ was used as an external reference. The redox potential level for the standard hydrogen electrode was calculated with the equation: $E_{NHE} = E_{SCE} + 0.24$ V.

### ROS generation in solution

The tetrahydrofuran solution of DPBF (10 mM), Ce6 (1 mM), **Ir1** (1 mM), **Ir-P1** (1 mg/mL), **DPP-PPy** (1 mg/mL), **Ir2** (1 mM), **Ir-P2** (1 mg/mL), and **QZL-DPP** (1 mM) were prepared as stock solution. MPdot (1 mg/mL) in water was prepared. The H$_2$O solutions of scavengers, Tiron (1 M) and t-butanol (1 M), were prepared for use.

The general procedure for ROS detection in DCM:

The tetrahydrofuran solution of PS was diluted in DCM to a final concentration of 2.80 µM and the Uv/Vis absorption spectra were recorded. 10 µL DPBF solution was added to the solution (final concentration was 50 µM) with its absorption spectra recorded. The mixture solution was irradiated with a 680 nm laser (5 mW·cm$^{-2}$) and its absorption spectra were recorded every 30 s. A time-dependent absorbance curve at 410 nm was drawn. The ROS quantum yield was calculated according to the following equation[60]:

$$\varnothing_{complex} = \varnothing_{Ce6} * \left(\frac{K_{complex}}{K_{Ce6}}\right) * \left(\frac{F_{Ce6}}{F_{complex}}\right) \quad (1)$$

where $K$ is the decomposition rate constants of DPBF calculated from the absorption decrease at 410 nm, in the presence of various PSs; $F = 1 - 10^{-OD}$, and OD is the absorbance at 680 nm for individual PS.

The general procedure for ROS scavenging in tetrahydrofuran/water (1 : 1, v/v):

MPdot was diluted in tetrahydrofuran/water (1 : 1) to 5.00 μg/mL (2.80 μM, calculated with one repeating unit), and the Uv/Vis absorption spectra were recorded. 10 μL DPBF and 10 μL scavenger solution were added to the mixture solution (final concentration was 50 μM for DPBF and 5 mM for scavenger) with its absorption spectra recorded. The mixture solution was then irradiated with a 680 nm laser (20 mW·cm$^{-2}$) and its absorption spectra were recorded every 30 s. A time-dependent absorbance curve at 410 nm was drawn.

### Transient absorption spectra

The Ultrafast System HELIOS TA spectrometer was used to capture the femtosecond transient absorption (TA) spectra of various samples. The laser source was provided by the Coherent Astrella−1K-F Ultrafast Ti: Sapphire Amplifier (800 nm, 1 kHz, <100 fs). Broadband probe pulses ranging from 450 to 775 nm were created by focusing a small fraction of the fundamental 800 nm laser pulses onto an $Al_2O_3$ plate. The 400-nm pump pulses were produced by doubling the fundamental 800-nm pulses using a BBO crystal.

In the time-profile fittings, we used following formula:

$$y = A + B_1 \exp\left(\frac{-t}{\tau_1}\right) + B_2 \exp\left(\frac{-t}{\tau_2}\right) + B_3 \exp\left(\frac{-t}{\tau_3}\right) \qquad (2)$$

where $B$ are relative amplitudes, and $\tau_1$, $\tau_2$ and $\tau_3$ represent the lifetimes of different photophysical pathways. In our case, the charge recombination is the last process and derived as $\tau_3$.

### Photoacoustic effect in solution

The photoacoustic imaging in solution was detected on the VEVO LAZR-X system (FUJIFILM VisualSonics, Toronto, Canada). For photoacoustic spectra, the photoacoustic signal was acquired at 680, 685, 700, ..., 745, 750 nm (100 ppm **MPdot1c** or **MPdot2c** in PBS, PA-Model(single), 100% power, 40-dB gain, 40-MHz frequency). For concentration-dependent photoacoustic spectra, the photoacoustic signal was acquired at 0, 6.25, 12.5, 25, and 50 ppm of **MPdot1c** or **MPdot2c** in PBS, (680 nm, PA-Model(single), 100% power, 40-dB gain, 40-MHz frequency). The intensity of the photoacoustic signal was determined by delineating the region of interest by the imaging system.

### Cell culture conditions

HeLa cells were incubated under 5.0% $CO_2$ and 21.0% $O_2$ at 37 °C in a humidified atmosphere in Dulbecco's Modified Eagle Medium (DMEM, Gibco BRL) with 10% (v/v) fetal bovine serum (FBS, Gibco BRL) and 100 μg/mL streptomycin and penicillin (Gibco BRL). The 4T1 cells were incubated under 5.0% $CO_2$ and 21.0% $O_2$ at 37 °C in a humidified atmosphere in RPMI 1640 Medium (Gibco BRL) with 10% (v/v) fetal bovine serum (FBS, Gibco BRL) and 100 μg/mL streptomycin and penicillin (Gibco BRL).

For hypoxic conditions, the $O_2$ level of the incubator was set to 1% with other conditions unchanged.

### In vitro ROS assay

50,000 HeLa cells were seeded in the 3.5 mL confocal microscopy dishes 16 hours in advance and divided into four groups with different treatments. MPdot coated with R8 in deionized water was diluted with DMEM to 20 μg/mL and used to incubate the cells in MPdot groups for 20 h. All cells were incubated with DHE or DHR 123 in DMEM without FBS for 0.5 h. The cells were further washed with PBS three times and the laser groups were irradiated with a 680 nm laser (40 mW·cm$^{-2}$) for 5 min. For the H$_2$DCFDA assay, the MPdot concentration was increased to 60 μg/mL and probe incubation to 1 hour, and irradiation time to 10 min. Then the PBS was replaced with fresh PBS and confocal fluorescence images were taken soon after the laser irradiation using Nikon A1R ($\lambda_{ex}$ = 488 nm, $\lambda_{em}$ = 515-535 nm).

For ROS assay by using flow cytometry, 50,000 HeLa cells were seeded in 12-well plates 16 h in advance and divided into four groups with different treatments. MPdot coated with R8 in deionized water was diluted with DMEM to 20 μg/mL and used to incubate the cells in MPdot groups for 20 h. All cells were then washed with PBS three times and incubated with DHE or DHR 123 (5 μM) in DMEM without FBS for 0.5 h. The cells were further washed with PBS three times and the cells in laser groups were irradiated with a 680 nm laser (40 mW·cm$^{-2}$) for 5 min. Then all cells were washed with fresh PBS, digested with trypsin, and collected for further detection and analysis using a flow cytometer. For the H$_2$DCFDA assay, the MPdot concentration was increased to 60 μg/mL and probe incubation to 1 h, and irradiation time to 10 min. For the scavenging experiments, 1,4-benzoquinone (1 mM) was added with the DHR 123 containing DMEM with other conditions unchanged.

### Cell apoptosis assay

50,000 HeLa cells were prepared in 12-well plates 16 h in advance and were randomly categorized into four groups for distinct treatments. The cells in the MPdot groups underwent treatment with MPdot (20 μg/mL) for 20 h. Subsequently, the cells in the laser groups were subjected to a 680 nm laser (100 mW·cm$^{-2}$) for 5 min. Following this, all cells were cultivated in fresh medium for an additional 8 h, then collected and stained with Annexin V-FITC (5 μL for 15 min) and PI (5 μL for 5 min). Finally, the cell population was quantified using flow cytometry with PE and FICT channel analyzed.

### Calcein-AM/PI assay

5000 HeLa cells were prepared in 96-well plates 12 h in advance. The cells were subsequently treated with MPdot at various concentrations of 0 μg/mL, 4 μg/mL, 8 μg/mL, and 12 μg/mL for 20 h. Following this, a 680 nm laser (100 mW·cm$^{-2}$) was used to irradiate cells in laser groups for 20 min. The cells were then cultivated for an additional 24 h in fresh DMEM. Subsequently, the cells were treated with a fresh medium containing Calcein-AM (10 μg/mL) and PI (5 μg/mL) for 20 min. Finally, the cells were gently rinsed with PBS for CLSM observation.

### Cell viability assay

5000 HeLa cells were prepared in 96-well plates 12 h in advance. The cells were then treated with MPdot at various concentrations ranging from 0 to 12 μg/mL for 20 h. The cells in the laser groups were exposed to a 680 nm laser (100 mW·cm$^{-2}$) for 20 minutes, while the cells in the dark group remained untreated. The medium for all groups was subsequently replaced with fresh, complete DMEM, and the cells were allowed to proliferate for an additional 24 h. This was succeeded by a replacement of the medium with a complete WST-8 containing DMEM medium (10%, v/v, Cell Counting Kit-8, C0039, Beyotime Biotechnology). Following 3 h, a Perkin Victor X5 microplate reader was used to measure the 450 nm absorbance of each well. Excel and GraphPad software (version 9.0.0) were used to determine the viability of the cells in each well and the IC50 value via nonlinear fitting.

### Western blot assay

Protein samples from HeLa cells that underwent different treatments (-10 μL) were subjected to polyacrylamide gel electrophoresis (SDS-PAGE, 10%) and subsequently transferred to nitrocellulose membranes. Bovine serum albumin (BSA) was used to block the membranes in a blocking buffer containing a 0.05% Tween, and this was incubated for 1.5 h at room temperature. Primary labeling was conducted by incubating the sample at 4 °C for 12 h with monoclonal antibody BcL-2 (Lot#B2117, sc-7382, Santa Cruz Biotechnology, Inc) at a 1:2000 dilution, using a blocking buffer. Rabbit anti-Mouse IgG Secondary Antibody (Catalog # 61-6520, Invitrogen) was used at a 1:1000 dilution as a secondary marker. Reactive protein bands were detected by exposing

the membranes to a luminol-hydrogen peroxide solution. Images were captured and analyzed with Image Bio-Rad Lab™ software.

## MPdot's cRGDyK peptide bioconjugation for in vivo therapy

To a rapidly stirred MPdot solution (2.8 mL, 1.54 mg/mL, with 40% PSMA), added fresh aqueous solution of N-(3-Dimethylaminopropyl)-N′-ethylcarbodiimide hydrochloride (12.4 μL, 7.2 mg/mL) and N-hydroxysuccinimide (4.97 μL, 10.8 mg/mL). The mixture was stirred at room temperature for 2 h, and the aqueous solution of cRGDyK (28.9 μL, 10 mg/mL, sequence: Cyclo(Arg-Gly-Asp-D-Tyr-Lys from Shanghai RoyoBiotech Co., Ltd) was added. The mixture was further stirred at room temperature for 24 h before purifying with a Sephadex® G-25 column to remove the free small molecules.

## In vivo photoacoustic imaging

Mice bearing 4T1 tumors were injected through the tail vein with cRGDyK-modified **MPdot1c** or **MPdot2c**, and unmodified **MPdot1** or **MPdot2** (400 pp, 100 μL, 2 mg kg⁻¹). For time-dependent MPdot accumulation at tumor photoacoustic imaging, the Single model of the Vevo LAZR was used. 0, 3, 7, 12, and 24 h after MPdot administration, the photoacoustic signal was acquired (680 nm, 100% power, 40-dB gain, 40-MHz frequency). The intensity of the photoacoustic signal in the tumor region was determined by delineating the region of interest by the imaging system.

The Oxy-Hemo imaging mode (40-dB gain, 10.00 mm depth, 12.08 mm width) was used to map the $O_2$ saturation(sO2), in which the 750 nm and 850 nm laser sources switched. sO2 and HbT in the epidermal area and tumor area were measured using an imaging system to outline the area of interest.

## In vivo anti-tumor therapy

4T1 tumor models were performed by subcutaneous injection of $5 \times 10^5$ 4T1 cells in 0.1 mL of PBS into the back of the mice. Seven days after model building, mice were randomly divided into four groups for different treatments. In **MPdotc** groups, a 100 μL solution of cRGDyK-modified **MPdotc** (**MPdot1c** or **MPdot2c**, respectively) at a concentration of 400 μg/mL was injected intravenously via tail (**MPdotc** dose: 2 mg kg⁻¹). The remaining mice were injected with 100 μL PBS solution instead. Twelve hours later, a 680 nm laser (300 mW·cm⁻²) was used to irradiate the tumor site for 10 min.

The mice in the laser-negative group were not treated. The same treatment was repeated on the 12th day. The tumor sizes were measured with calipers calculated using the following formula:

$$V = \frac{\pi}{6} * A * B^2 \qquad (3)$$

Where A is the long diameter of the tumor and B is the shot diameter.

On the 17th day, all the mice were sacrificed with tumors and organs dissected.

## Reporting summary

Further information on research design is available in the Nature Portfolio Reporting Summary linked to this article.

## Data availability

The data that supports the findings of this study can be found in the manuscript, and its Supplementary Information, or are available from the corresponding author upon request.

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

## Acknowledgements

This work was supported by (1) the Science and Technology Development Fund received by X.J.Z., Macau SAR (File No.: 0114/2019/A2, 0085/2020/A2); (2) Shenzhen Science and Technology Program received by C.F. Wu (Grant No. KQTD20170810111314625, JCYJ20210324115807021); (3) the Research Grant of the University of Macau received by X.J.Z.

(under grant No.: MYRG2020-00130-FHS, MYRG2022-00036-FHS); (4) Guangdong Basic and Applied Basic Research Foundation received by X.J.Z. (2022A1515010616, and 2023A1515012524). We thank the core facilities in the Faculty of Health Sciences, especially the drug development core, animal research core, bioimaging, and stem cell core for their excellent services.

## Author contributions

Z.Z. and X.J.Z. conceived the idea and designed the experiments. Z.Z. conducted the synthesis, cell, and animal experiments. Z.X.W. conducted the EPR characterization. J.T.G. and Z.Y. conducted the PA and PTT characterization in solution and PA characterization in vivo with data processing. B.Z.W., G.W., and G.C.X. conducted the transient absorption experiments and data processing. J.X.L. provided support in synthesis and animal experiments. C.F. Wang and L.Q.Z. provided support in cell and animal experiments. C.F. Wu acquired the funds and studied the optical properties. X.J.Z. acquired the funds and supervised the project. Z.Z and X.J.Z. wrote and revised the manuscript. All authors analyzed the data, discussed the results, and commented on the manuscript.

## Competing interests

The authors declare no competing interests.
