## [Peer Review File · Nature Communications]

Reviewers' Comments:

Reviewer #1:

Remarks to the Author:

In this manuscript, the authors present an approach using narrow bandgap metallopolymer to develop Type-I photosensitizers with strong NIR absorption. Two metallopolymers, Ir-P1 and Ir-P2, were designed and their photophysical and photosensitive properties were systematically studied. The results demonstrated that the properties of metallopolymers were significantly improved compared to their monomers. Notably, these photosensitizers can effectively generate superoxide radicals through Type-I photodynamic pathway, and their tumor suppression effect is almost unaffected by hypoxia. Overall, this is an interesting work, but there are still some issues that need to be addressed.

1. In this manuscript, the authors did not explicitly explain why the efficiency of ROS production was enhanced, was it due to the enhanced absorption by the photosensitizers? If so, why wasn't the efficiency of singlet oxygen generation enhanced? Please comment on this.
2. In the ESR test, the authors detected the characteristic signal of the adduct of hydroxyl radicals with DMPO, but there was almost no signal of superoxide radicals. The authors attributed this to the thermal decomposition of the addition product of $O_2-\bullet$ with DMPO. Would it be possible to use a more stable spin-trap agent (e.g. BMPO) to detect the addition product of superoxide radicals?
3. The authors describe that DBPF is widely used as an indicator for ROS detection. However, as far as I know, DPBF is usually used to detect the production of singlet oxygen, while DCFH (2',7'-dichlorodihydrofluorescein) is commonly used to detect ROS.
4. Regarding the use of Rhodamine 123 for selectively detecting $O_2-\bullet$, I think it needs to be further verified. Some literature suggests that Rhodamine 123 is not selective for superoxide radicals, as it can also be oxidized by other ROS.
5. DHE (dihydroethidium) is generally considered a good specific probe for detecting $O_2-\bullet$. It is recommended to supplement the experiment with the use of DHE to detect the generation of $O_2-\bullet$ (refer to Chem. Sci., 2022, 13, 5951-5956; J. Am. Chem. Soc. 2023, 145, 4081).
6. The authors describe that H2DCFDA hardly reacts with superoxide radicals. However, it is generally reported in the literature that H2DCFDA can enter cells and hydrolyze into DCFH, which can react with various types of ROS, including singlet oxygen and superoxide radicals (Free Radical Res. 2010, 44, 587-604). Moreover, current imaging results of ROS can only indicate whether ROS is produced, but cannot compare the amount of ROS produced. Please consider the relevant description carefully.

Reviewer #2:

Remarks to the Author:

This manuscript describes a novel strategy to develop a deep-red light-driven type I polymer photosensitizer with an Ir complex and low-band gap polymer. I enjoyed reading the manuscript and understood the potential of this strategy. However, there are some unclear points in the strategy, and I recommend a major revision before acceptance in Nature Communications. The points I am concerned about are shown below:

1. The authors do not discuss why the metallopolymers show type-I photosensitization. If the authors would like to emphasize that this strategy is new, they should elucidate the mechanism. In the current version, I could not understand what the new strategy for the development of type I photosensitizers is, and I suspect that just mixing a metal complex and a narrow band gap polymer may not give a type I photosensitizer.
2. The redox potential levels of the polymers are an important factor for electron transfer to oxygen. The experiments should be performed in the revised manuscript.
3. Do the metallopolymers combined with R-8 peptide and PDMS polymer (i.e., MPdot1r and MPdot1c) exhibit the same photosensitization properties as MP-dots?
4. On page 15, the authors describe that a photothermal effect can be observed. However, there is no data to support this claim.
5. I am confused about the superoxide detection experiments using rhodamine 123, because it is an inherently emissive molecule. I think that the authors may use dihydrorhodamine 123 (DHR 123) instead. DHR 123 indeed gives rhodamine 123 after reacting with superoxide, but DHR 123 also

reacts with ROS other than superoxide. Therefore, the superoxide detection experiments are doubtful. More reliable reagents such as WST-1 would be better for superoxide detection.

6. In the in vivo experiments, why are two-cycle treatments needed? Generally, an effective PS for PDT requires only a one-shot treatment.

Reviewer #3:

Remarks to the Author:

This paper "Metallopolymer Strategy to Explore Hypoxic Active Narrow-Bandgap Photosensitizers for Effective Cancer Photodynamic Therapy" by Zhang et al. reports the synthesis and application of Ir(III) complexes covalently coupled or directly included in the conjugated polymer backbones. The small molecular Ir(III) compounds have type I PDT ability, but their absorption wavelengths are short (in the blue range) and the absorption coefficient is low, thus exhibiting low ROS generating efficiency. After being coupled or included in the conjugated polymer, the obtained macromolecular Ir(III) complexes (Ir-P1 and Ir-P2) showed red-shifted absorption wavelength and enhanced absorption coefficient. The authors demonstrated the ROS production properties of the corresponding nanoparticles (MPdot1 and MPdot2) in cells and proved their treatment effect in the mice models of tumors. Overall, this paper is more suitable for some specialized journals, but not Nature Communications. Here are two major issues and one minor issue of this paper:

Firstly, after reading this paper, it is not persuasive that the novelty of this work would be enough for Nature Communications. This is not the first type I photosensitizer based on iridium compounds, since there are already quite a few examples of iridium complexes reported for type I PDT. This is also not the first paper that utilizes the strategy of combining iridium compound and conjugated polymers for PDT. The reported macromolecular Ir(III) complexes (Ir-P1 and Ir-P2) do have special advantages in terms of some detailed parameters, e.g. longer absorption wavelength and larger absorbance. But this is more like an incremental progress in that specific field rather than a revolutionary scientific finding.

Secondly, the data of the animal experiments cannot support the unique advantages of the reported Ir-based photosensitizers. The special part of the reported Ir-based photosensitizers (MPdot1 and MPdot2) is that they have type I PDT ability instead of traditional type II and thus they would work well under hypoxic condition. However, in the animal experiments, the authors just demonstrate that the laser + MPdot1 group showed better treatment effect than the laser only or MPdot1 only groups. Of course, the laser + photosensitizer group will work better than the laser only or photosensitizer only group. Many reported studies based on type II PDT have demonstrated very similar results like this. In this paper, there is no evidence that the tumors were hypoxic, conventional type II PDT did not work well for this kind of tumors, and thus the reported type I PDT method must be used. The design of the current animal experiment is way too simple and the result cannot reflect what is really new for the reported Ir-based photosensitizers.

In addition, there is a minor issue: two Ir-based photosensitizers (MPdot1 and MPdot2) are reported in this paper. It seems that they always exhibit similar behaviors, including similar size, similar ROS production in solution and similar performance in the cell studies. And then in the animal experiment, only MPdot1 was tested, while MPdot2 was forgotten. If the authors report two nanoparticles in a paper, the readers may expect that these two nanoparticles would have some difference and be applied for different circumstances. If the second one is totally not needed for further applications, the readers may doubt that why the authors introduce two photosensitizers at the beginning.

Responses to Reviewers' Comments

Reviewer #1 (Remarks to the Author):

In this manuscript, the authors present an approach using narrow bandgap metallopolymer to develop Type-I photosensitizers with strong NIR absorption. Two metallopolymers, Ir-P1 and Ir-P2, were designed and their photophysical and photosensitive properties were systematically studied. The results demonstrated that the properties of metallopolymers were significant improved compared to their monomers. Notably, these photosensitizers can effectively generate superoxide radicals through Type-I photodynamic pathway, and their tumor suppression effect is almost unaffected by hypoxia. Overall, this is an interesting work, but there are still some issues that need to be addressed.

Q1. *In this manuscript, the authors did not explicitly explain why the efficiency of ROS production was enhanced, was it due to the enhanced absorption by the photosensitizers? If so, why wasn't the efficiency of singlet oxygen generation enhanced? Please comment on this.*

Response: We thank the reviewer for the valuable comments. In this work, the metallopolymerization strategy addresses the absorption problem mainly in two aspects: the wavelength and absorption intensity. The ROS production improvement is highly related to enhanced absorption. And according to the energy band hybridization hypothesis, the photosensitization can also be enhanced (*Chem*, **2018**, 4, 1937).

We also thank the reviewer for the question “*why wasn't the efficiency of singlet oxygen generation enhanced?*”. In our work, we observed that the cyclometalated iridium complexes with C[^]N-ligands (phenylpyridine, for example) tend to show Type I photochemical pathways while analogues from N[^]N-ligands (bipyridine) exhibit Type II photoreaction as confirmed by ESR test. This phenomenon was also observed in Ru(II) complexes and it was explained that the electron-donating cyclometalated C[^]N-ligands can elevate the energy level of the d π (Ru) orbital and make the oxidation potential cathodic shift, providing a possibility for type I PSs by electron transfer (*Chem. Sci.*, **2018**, 9, 502; *Eur. J. Inorg. Chem.*, **2012**, 25, 4004). In our work, the redox levels of two polymers **Ir-P1** and **Ir-P2** were measured in cyclic voltammetry, which indicated that these polymers were thermodynamically feasible for Type I photochemical reactions. As a result, the Type I photochemistry is highly possibly more favorable due to the Type I “Iridium catalytic center” and a suitable redox level.

Q2. *In the ESR test, the authors detected the characteristic signal of the adduct of hydroxyl radicals with DMPO, but there was almost no signal of superoxide radicals. The authors attributed this to the thermal decomposition of the addition product of O₂-• with DMPO.*

Would it be possible to use a more stable spin-trap agent (e.g. BMPO) to detect the addition product of superoxide radicals?

Response: We thank the reviewer for valuable suggestion. BMPO was used and the signal for BMPO-OOH, the product of $O_2^{\cdot-}$, was observed. We have updated the results in revised manuscript.

Figure. The EPR data was obtained from an aqueous solution (DMSO:H₂O = 1:9, v/v) containing **BMPO**. Experimental conditions: [MPdot1] = [MPdot2] = 0.5 mM, [BMPO] = 100 mM; irradiation: 680 nm laser (800 mW·cm⁻²).

Q3. The authors describe that DBPF is widely used as an indicator for ROS detection. However, as far as I know, DPBF is usually used to detect the production of singlet oxygen, while DCFH (2',7'-dichlorodihydrofluorescein) is commonly used to detect ROS.

Response: We thank the reviewer for the valuable comment. We have also supplemented ROS assay with DCFH (please see response to question Q6) according to your suggestion. But here we would like to explain the use of DBPF first. In fact, the use of DPBF as a probe for superoxide radical in phospholipid liposomal membranes can be dated back to 1999 (*Biochimica. et Biophysica. Acta.*, **1999**, 1421, 131). It is reported that DPBF also reacts with hydroxy, alkyloxy or alkylperoxy radicals, peroxyxynitrite anion, superoxide anion, hydrogen peroxide, and hypochlorite anion. (Chmurzyński et al, *Free Radical Res.*, **2017**, 51, 38). In this review, the probes for detecting ROS have been summarized (*J Biochem. Biophys. Methods*, **2005**, 65, 45). Additionally, DPBF was usually used to quasi-evaluation of ROS generating efficiency, and also for Type-I ROS generation evaluation (*Chem. Sci.*, **2018**, 9, 502; *Chem. Mater.*, **2019** 31, 3313; *Nat Commun.*, **2022**, 13, 2225). And based on our ROS-type determination tests in solution with **Tirons** as superoxide radical scavengers, it is sufficient to prove that the use of DPBF as an indicator for ROS detection is proper in our article. To acquire the more quantitative evaluation of the ROS generation ability, Ce6 as a reference

photosensitizer and DPBF as the probe instead of DCFH, were used consequently.

We have supplemented the detection of ROS in solution using freshly prepared DCFH (by hydrolyzing H₂DCFDA) and in cells using H₂DCFDA (please see response to question Q6).

Q4. Regarding the use of Rhodamine 123 for selectively detecting O₂^{-•}, I think it needs to be further verified. Some literature suggests that Rhodamine 123 is not selective for superoxide radicals, as it can also be oxidized by other ROS.

Response: We agree with the reviewer about this point. Dihydrorhodamine **123** (**DHR 123**) can be oxidized by other ROS such as HOCl and ONOO[•]. But in the ROS type determining section, as shown in **Figure 3(e)**, where only **DHR 123** and MPdot were added, without any other reagents, the green fluorescence was observed after irradiation. Additionally, in solution detection experiment, **DHR 123** is conventionally used for its convenience (*J. Am. Chem. Soc.* **2018**, *140*, 14851; *Chem. Mater.*, **2019** *31*, 3313; *Adv. Mater.* **2022**, *34*, 2108146; *J. Am. Chem. Soc.* **2016**, *138*, 10968). It would be convincing to use **DHR 123** to preliminarily to detect and support O₂^{-•} generation in solution. However, according to your suggestions in question Q5, we have also detected O₂^{-•} using DHE (please see response to Q5) below.

*Q5. DHE (dihydroethidium) is generally considered a good specific probe for detecting O₂^{-•}. It is recommended to supplement the experiment with the use of DHE to detect the generation of O₂^{-•} (refer to *Chem. Sci.*, 2022, *13*, 5951-5956; *J. Am. Chem. Soc.* 2023, *145*, 4081).*

Response: We thank the review for this suggestion. We have supplemented a series of experiments using **DHE** for detecting O₂^{-•} and the generation of O₂^{-•} was confirmed by flow cytometry, confocal cell imaging, and tumor imaging.

Figure. ROS assay with **DHE**. Experimental conditions: [**MPdot1r**] = 20 μg/mL with preincubation for 20 hours, [**DHE**] = 5 μM with incubation for 30 mins, 680 nm laser irradiation (40 mW·cm⁻²) for 5 mins. (a₁-a₃). Laser-/MPdot1r-; (b₁-b₃). Laser+/MPdot1r-; (c₁-c₃). Laser-/MPdot1r+; (d₁-d₃). Laser+/MPdot1r+; (e). Flow cytometry for O₂^{•-} detection with **DHE**.

Figure. ROS assay with DHE. Experimental conditions: [MPdot2r] = 20 μg/mL with preincubation for 20 hours, [DHE] = 5 μM with incubation for 30 mins, 680 nm laser irradiation (40 mW·cm⁻²) for 5 mins. (a₁-a₃). Laser-/MPdot2r-; (b₁-b₃). Laser+/MPdot2r-; (c₁-c₃). Laser-/MPdot2r+; (d₁-d₃). Laser+/MPdot2r+; (e). Flow cytometry for O₂^{-•} detection with DHE.

Figure. *In vivo* O₂^{-•} detection with DHE. The mice in MPdot groups were dosed with MPdot1c and MPdot2c (100 μL, 400 ppm) via tail intravenous injection 12 hours before the administration. DHE in PBS (1mM) was intratumorally injected for all mice 30 mins before laser irradiation.

Q6. The authors describe that H₂DCFDA hardly reacts with superoxide radicals. However, it is generally reported in the literature that H₂DCFDA can enter cells and hydrolyze into DCFH, which can react with various types of ROS, including singlet oxygen and superoxide radicals (*Free Radical Res.* 2010, 44, 587–604). Moreover, current imaging results of ROS can only indicate whether ROS is produced but cannot compare the amount of ROS produced. Please consider the relevant description carefully.

Response: We seriously re-considered this issue and conducted the ROS detection experiment in solution with freshly prepared DCFH (by hydrolyzing H₂DCFDA, pre-treated with 0.01M NaOH solution, *Anal. Chem.* 2017, 89, 7, 3853; *Aerosol Science and Technology*, 2014, 12, 1276). In the presence of 680 nm laser, green fluorescence of DCF was observed upon activating **MPdot1** or **MPdot2**. Additionally, the amount of ROS generated in cells can be compared based on flow cytometry results, where the higher portion of cells with higher fluorescence indicated the greater amount of ROS generated. And according to these results, **MPdot1** and **MPdot2** both showed excellent ROS generation in cells and **MPdot1** exhibited even a little better performance, which consisted with ROS generation results in solution.

Figure. The fluorescence spectra for ROS probes in water. Experimental conditions: [H₂DCF] = 20 μM, [MPdot1] = [MPdot2] = 2.8 μM with a 680 nm laser (20 mW·cm⁻²). (a). **MPdot1**; (b). **MPdot2**.

Figure. ROS assay with H_2DCFDA . Experimental conditions: [MPdot1r] = 60 $\mu\text{g}/\text{mL}$ with preincubation for 20 hours, [H_2DCFDA] = 5 μM with incubation for 30 mins, 680 nm laser irradiation (40 $\text{mW}\cdot\text{cm}^{-2}$) for 10 mins. (a₁-a₃). Laser-/MPdot1r-; (b₁-b₃). Laser+/MPdot1r-; (c₁-c₃). Laser-/MPdot1r+; (d₁-d₃). Laser+/MPdot1r+; (e). Flow cytometry ROS detection with H_2DCFDA .

Figure. ROS assay with H_2DCFDA . Experimental conditions: [MPdot2r] = 60 $\mu\text{g}/\text{mL}$ with preincubation for 20 hours, [H_2DCFDA] = 5 μM with incubation for 30 mins, 680 nm laser irradiation ($40 \text{ mW} \cdot \text{cm}^{-2}$) for 10 mins. (a₁-a₃). Laser-/MPdot2r-; (b₁-b₃). Laser+/MPdot2r-; (c₁-c₃). Laser-/MPdot2r+; (d₁-d₃). Laser+/MPdot2r+; (e). Flow cytometry ROS detection with H_2DCFDA .

Reviewer #2 (Remarks to the Author):

This manuscript describes a novel strategy to develop a deep-red light-driven type I polymer photosensitizer with an Ir complex and low-band gap polymer. I enjoyed reading the manuscript and understood the potential of this strategy. However, there are some unclear points in the strategy, and I recommend a major revision before acceptance in Nature Communications. The points I am concerned about are shown below:

***Q1.** The authors do not discuss why the metallopolymers show type-I photosensitization. If the authors would like to emphasize that this strategy is new, they should elucidate the mechanism. In the current version, I could not understand what the new strategy for the development of type I photosensitizers is, and I suspect that just mixing a metal complex and a narrow band gap polymer may not give a type I photosensitizer.*

Response: We thank the review for the valuable comments, which are very useful for us to improve this work. We agree with you that simply mixing a metal complex and semiconducting polymer cannot guarantee efficient type-I ROS generation. In terms of the origin of Type-I photosensitization, we observed that the cyclometalated iridium complexes with C^N-ligands (phenylpyridine, for example), tend to show Type I photochemical pathways while analogues from N^N-ligands (bipyridine) exhibit Type II photoreaction as confirmed by ESR test. This phenomenon was also observed in Ru(II) complexes and it was explained that the electron-donating cyclometalated C^N-ligands can elevate the energy level of the d π (Ru) orbital and make the oxidation potential cathodic shift, providing a possibility for type I PSs by electron transfer (*Chem. Sci.*, **2018**, *9*, 502; *Eur. J. Inorg. Chem.*, **2012**, *25*, 4004). In our work, after screening two type-I iridium complexes, we engineered them with semiconducting polymers to get metallopolymers, which keep the type-I photochemical pathways and also fully utilize the intense long-wavelength absorbance of low-bandgap polymers. The redox levels of two polymers **Ir-P1** and **Ir-P2** were measured in cyclic voltammetry, which indicated that these polymers were thermodynamically feasible for Type I photochemical reactions. As a result, the Type I photochemistry is highly possibly more favorable due to the Type-I “Iridium catalytic center” and a suitable redox level.

In this work, we aimed to prepare PSs for practical therapy with three basic criteria: high-performance (high ROS generation efficiency), Type I (suit to hypoxia), and bio-friendly therapeutic window (long-wavelength activation). By engineering Type-I iridium(III) complexes, a facile metallopolymerization strategy was developed to promote its Type-I ROS generation and address the absorption problem at the same time. This strategy shows its generality and potential in preparing a series of photosensitizers. In fact, we synthesized metallopolymers (**Ir-F8**, **Ir-TBT**, **Ir-P1**) by combining iridium unit with blue-, green-, and deep

red-emitting polymers. And Type-I ROS generations were observed in electron paramagnetic resonance (EPR) test for all **MPdots**, which suited to the Type I criteria. These facts showed that our strategy is of great value to design new type of polymeric PSs. More detailed and interesting study can be conducted based on our strategy and bring deeper understanding of the photochemical mechanism. We did not discuss **Ir-F8**, **Ir-TBT** in the first version of manuscript due to their relatively short absorbance wavelengths, which violated the bio-friendly therapeutic window criteria. We have added some discussions about the mechanism in revised manuscript.

Figure. Chemical structure of Ir-F8 (a) and Ir-TBT (b). and the EPR data (c, d) was obtained from an aqueous solution (DMSO:H₂O = 1:9, v/v) containing BMPO. Experimental conditions: [MPdotF8] = [MPdotTBT] = 0.5 mM, [BMPO] = 100 mM; irradiation: white LED light (50 mW·cm⁻²).

Q2. The redox potential levels of the polymers are an important factor for electron transfer to oxygen. The experiments should be performed in the revised manuscript.

Response: We thank the reviewer for this valuable suggestion. The cyclic voltammetry for metallopolymers was conducted indicating that they are thermodynamically feasible to generate O₂^{•-}. We have added the results in the revised manuscript.

Figure. Cyclic voltammetry test performed on HY 1550A mini electrochemical analyzer. Experimental conditions: Scan rate $100 \text{ mV} \cdot \text{s}^{-1}$ in anhydrous dichloromethane, $[\text{Ir-P1}] = [\text{Ir-P2}] = 2 \text{ mg} \cdot \text{mL}^{-1}$, $[\text{Bu}_4\text{NPF}_6] = 0.2 \text{ M}$. (a). Cyclic voltammograms for **Ir-P1**, red line; for **Ir-P2**, blue line, and **Fc**, black line. (b). Redox potential diagram of **Ir-P1**, red line and **Ir-P2**, blue line.

Q3. Do the metallopolymers combined with R-8 peptide and PDMS polymer (i.e., *MPdot1r* and *MPdot1c*) exhibit the same photosensitization properties as MP-dots?

Response: The post-modification of the MPdot did not alter the photosensitization much, as shown in the **Figure** below.

Figure. Absorption spectra for **DPBF** in tetrahydrofuran/water (1:1) solution. Experimental conditions: $[\text{DPBF}] = 50 \text{ } \mu\text{M}$, $[\text{MPdot}] = 2.8 \text{ } \mu\text{M}$, with a 680 nm laser ($20 \text{ mW} \cdot \text{cm}^{-2}$) and recorded every 30 s. (a). **MPdot1**; (b). **MPdot1r**; (c). **MPdot1c**; (d). **MPdot2**; (e). **MPdot2r**; (f). **MPdot2c**.

Q4. On page 15, the authors describe that a photothermal effect can be observed. However, there is no data to support this claim.

Response: We have conducted a concentration-dependent photothermal effect experiment for **MPdot1**, and the results have been presented in Figure below. And please be noted that a much higher concentration (~ 900 ppm) was used for EPR test, and the PTT effect would be more obvious. This photothermal effect may promote the transformation of superoxide to hydroxyl radical.

Figure. The concentration-dependent photothermal effect for MPdot1 in the presence of 680nm laser ($1.2\text{W}\cdot\text{cm}^{-2}$)

Q5. I am confused about the superoxide detection experiments using rhodamine 123, because it is an inherently emissive molecule. I think that the authors may use dihydrorhodamine 123 (DHR 123) instead. DHR 123 indeed gives rhodamine 123 after reacting with superoxide, but DHR 123 also reacts with ROS other than superoxide. Therefore, the superoxide detection experiments are doubtful. More reliable reagents such as WST-1 would be better for superoxide detection.

Response: Sorry for the typo. We used dihydrorhodamine 123 (DHR 123) for experiment but wrote it as rhodamine 123 by mistake. We have also supplemented the detection of $\text{O}_2^{\cdot-}$ with WST-1 according to your suggestion. Firstly, we would like to discuss the detection of $\text{O}_2^{\cdot-}$ with **dihydrorhodamine 123 (DHR 123)**. DHR123 can be oxidized by other ROS such as HOCl and ONOO \cdot . But in the ROS type determining section in water, as shown in **Figure 3(e)**, where only **DHR 123** and MPdot were added, without any other reagents, the green fluorescence was observed after irradiation. Additionally, in solution detection experiment, **DHR 123** is conventionally used for its convenience (*J. Am. Chem. Soc.* **2018**, *140*, 14851; *Chem. Mater.*, **2019** *31*, 3313; *Adv. Mater.* **2022**, *34*, 2108146; *J. Am. Chem. Soc.* **2016**, *138*, 10968). It would be convincing to use DHR 123 to preliminarily detect $\text{O}_2^{\cdot-}$ in solution.

To further confirm the generation of $\text{O}_2^{\cdot-}$, we have conducted $\text{O}_2^{\cdot-}$ detection with WST-1

according to your valuable suggestion, and the results were presented in **Figure** below. Upon the irradiation of 680 nm laser, WST-1 was slowly evolved, indicating the generation of O_2^- . To better show the increment in **MPdot2** case, absorption at 411 instead of conventionally at 435 nm was monitored.

Figure. The absorption spectra of **WST-1** in aqueous solutions, pH = 8.0. Experimental conditions: $[WST-1] = 50 \mu M$, $[MPdot1] = [MPdot2] = 14.0 \mu M$ with a 680 nm laser ($20 mW \cdot cm^{-2}$) and recorded every 1 min. (a). no photosensitizer; (b). **MPdot1**; (c). **MPdot2**. And the incremental absorption at 411 nm: (d). no photosensitizer; (e). **MPdot1**; (f). **MPdot2**.

Q6. *In the in vivo experiments, why are two-cycle treatments needed? Generally, an effective PS for PDT requires only a one-shot treatment.*

Response: We agree with the reviewer about this point. However, to ensure the maximal inhibition or total eradication of tumor, several cycles may be needed even for effective PS, since in many reported works, small tumor was still harvested after one-cycle treatment which may proliferate once again. But we take your advice seriously and performed the one-cycle treatment (tail vein injection). MPdots were dosed via tail vein injection, mice in MPdot group were irradiated after 12 hours, and further observed for 6 days. The mice were sacrificed and dissected with tumor collected. It can be confirmed just for one-cycle treatment, the tumor can be inhibited efficiently. Here, to avoid wasting more mice, we would like to keep using the results of two-cycle treatment in the article.

Figure. (a). Tumor volume was measured at indicated day with a caliper and calculated using the following equation: $V = \frac{\pi}{6} \times A \times B^2$, A is long diameter and B is short diameters of the tumor. (b). Tumor weight measured after dissection. (c). Mice photo and (d). Tumor photo for mice in each group. *****, $P < 0.0001$

Reviewer #3 (Remarks to the Author):

This paper “Metallopolymer Strategy to Explore Hypoxic Active Narrow-Bandgap Photosensitizers for Effective Cancer Photodynamic Therapy” by Zhang et al. reports the synthesis and application of Ir(III) complexes covalently coupled or directly included in the conjugated polymer backbones. The small molecular Ir(III) compounds have type I PDT ability, but their absorption wavelengths are short (in the blue range) and the absorption coefficient is low, thus exhibiting low ROS generating efficiency. After being coupled or included in the conjugated polymer, the obtained macromolecular Ir(III) complexes (Ir-P1 and Ir-P2) showed red-shifted absorption wavelength and enhanced absorption coefficient. The authors demonstrated the ROS production properties of the corresponding nanoparticles (MPdot1 and MPdot2) in cells and proved their treatment effect in the mice models of tumors. Overall, this paper is more suitable for some specialized journals, but not Nature Communications. Here are two major issues and one minor issue of this paper:

Q1. Firstly, after reading this paper, it is not persuasive that the novelty of this work would be enough for Nature Communications. This is not the first type I photosensitizer based on iridium compounds, since there are already quite a few examples of iridium complexes reported for type I PDT. This is also not the first paper that utilizes the strategy of combining iridium compound and conjugated polymers for PDT. The reported macromolecular Ir(III) complexes (Ir-P1 and Ir-P2) do have special advantages in terms of some detailed parameters, e.g. longer absorption wavelength and larger absorbance. But this is more like an incremental progress in that specific field rather than a revolutionary scientific finding.

Response: We thank the reviewer for the valuable comments. In this work, we aimed to prepare PSs for practical therapy with three basic criteria: high-performance (high ROS generation efficiency), Type I (suit to hypoxia), and bio-friendly therapeutic window (long-wavelength activation). By engineering a Type-I iridium(III) complex, a very easy metallopolymerization strategy was developed to promote its Type-I ROS generation and address the absorption problem at the same time. This strategy also shows its generality. In fact, we synthesized metallopolymers (**Ir-F8**, **Ir-TBT**, **Ir-P1**) by combining iridium unit with blue-, green-, and deep red-emitting polymers. And Type-I ROS generations were observed in electron paramagnetic resonance (EPR) test for all **MPdots**, which suited to the Type I criteria. As an example, **Ir-F8** exhibited brilliant ROS generation under shine with 473nm laser, times higher than that of reference **Ce6**, which met the high-performance criteria. We did not discuss **Ir-F8** and **Ir-TBT** in last version of manuscript due to their short absorption wavelengths. But these facts showed our strategy is of great value in designing new type of polymeric PSs (wider significance in

terms of a potentially more generalizable approach). We have added some discussion about this point in the revised manuscript. In short, our strategy addresses the high-performance ROS generation, issue of absorption wavelength and type-I photochemistry at one time. We believe that our design strategy brings new view for the development of organic PSs.

Figure. Chemical structure of Ir-F8 (a) and Ir-TBT (b), and the EPR data (c, d) was obtained from an aqueous solution (DMSO:H₂O = 1:9, v/v) containing BMPO. Experimental conditions: [MPdotF8] = [MPdotTBT] = 0.5 mM, [BMPO] = 100 mM; irradiation: white LED light (50 mW·cm⁻²).

Q2. Secondly, the data of the animal experiments cannot support the unique advantages of the reported Ir-based photosensitizers. The special part of the reported Ir-based photosensitizers (MPdot1 and MPdot2) is that they have type I PDT ability instead of traditional type II and thus they would work well under hypoxic condition. However, in the animal experiments, the authors just demonstrate that the laser + MPdot1 group showed better treatment effect than the laser only or MPdot1 only groups. Of course, the laser + photosensitizer group will work better than the laser only or photosensitizer only group. Many reported studies based on type II PDT have demonstrated very similar results like this. In this paper, there is no evidence that the tumors were hypoxic, conventional type II PDT did not work well for this kind of tumors, and thus the reported type I PDT method must be used. The design of the current animal experiment is way too simple and the result cannot reflect what is really new for the reported Ir-based photosensitizers.

Response: We thank the reviewer for these comments. We agree with the reviewer that the laser + photosensitizer group will work better than the laser only or photosensitizer only group. However, for research in this field, the results from laser only or photosensitizer only groups

are usually needed to compare with laser + photosensitizer group. According to the suggestions from the reviewer and Editor, we detected the tumor hypoxia *in vivo*. We used commercially available Hypoxyprobe HP6-Kit following standard procedure. As shown in the figure below, the green fluorescence strongly supports the hypoxia feature in solid tumor. And the key part of our article is revealing design guidelines for high-performance Type I photosensitizers with bio-friendly operation window. The animal experiments were simple but sufficient to prove the excellent therapeutic effect *in vivo* of the newly developed MPdots.

Figure. Hypoxic detection in tumor tissue with commercially available HypoxyprobeTM-1 Green Kit. (a). FITC channel; (b). DAPI; (c). Merged channel.

Q3. In addition, there is a minor issue: two Ir-based photosensitizers (MPdot1 and MPdot2) are reported in this paper. It seems that they always exhibit similar behaviors, including similar size, similar ROS production in solution and similar performance in the cell studies. And then in the animal experiment, only MPdot1 was tested, while MPdot2 was forgotten. If the authors report two nanoparticles in a paper, the readers may expect that these two nanoparticles would have some difference and be applied for different circumstances. If the second one is totally not needed for further applications, the readers may doubt that why the authors introduce two photosensitizers at the beginning.

Response: We thank the reviewer for the comments. Indeed, MPdot1 and MPdots 2 show similar behaviors such as sizes, colloidal stability, and some other properties, because Pdot technique shows good reproducibility. But the photophysical and photochemical properties also depend on the polymers' chemical structures. For example, we also combined Iridium with blue-emitting, green-emitting, and red-emitting polymers, which show quite different photophysical properties. The aim of this work is to develop potentially generalizable approach to metallopolymers. Other researchers may reference our method to synthesize metallopolymers and nanoparticle based on the availability of starting materials and their applications (not limited to photodynamic therapy). We added more discussion about this point in revised manuscript. However, we take your comments seriously and conducted the *in vivo* therapeutic evaluation of **MPdot2**. Firstly, *in vivo* O₂^{•-} detection with **DHE** confirmed the type-

I photodynamic therapy. It was demonstrated that **MPdot2** also have excellent therapeutic effect and negligible dark toxicity for major organs. We have added the results in supporting information.

Figure. *In vivo* $O_2^{\cdot -}$ detection with DHE. The mice in MPdot groups were dosed with **MPdot1c** and **MPdot2c** (100 μ L, 400 ppm) via tail intravenous injection 12 hours before the administration. **DHE** in PBS (1mM) was intratumorally injected for all mice 30 mins before laser irradiation.

Figure. (a). Tumor volume was measured at indicated day with a caliper and calculated using the following equation: $V = \frac{\pi}{6} \times A \times B^2$, A is long diameter and B is short diameters of the tumor. (b). Tumor weight measured after dissection. (c). Mice photo and (d). Tumor photo for mice in each group. *****, $P < 0.0001$

Reviewers' Comments:

Reviewer #1:

Remarks to the Author:

I think the manuscript can be published in its current version.

Reviewer #2:

Remarks to the Author:

The revised manuscript is indeed improved compared to the previous one. However, the strategy employed in this work may not possess the necessary novelty required for publication in Nature Communications. The electron transfer between the metal and ligand for type I photosensitization and metallo polymerization are well-established and widely used conventional strategies.

Therefore, the manuscript would be more suitable for submission to specialized journals in the field.

Reviewer #3:

Remarks to the Author:

The authors have added the data of hypoxia detection in tumor tissue with commercially available HpoxyprobeTM-1 Green Kit, which is helpful to support their conclusion. Here are the remaining issues to be addressed:

1) The hypoxia staining images by HpoxyprobeTM-1 Green Kit is shown in the response letter, but it seems that these images are not included in main text or supplementary information. Besides, I see the authors have performed photoacoustic imaging of the tumor tissues over time. I don't understand why they didn't characterize the tumor oxygenation state by photoacoustic imaging, which can be easily performed by analyzing the signals from the endogenous hemoglobin. This will be much better than the immunofluorescence staining method, because the PA imaging can demonstrate the hypoxia state of the whole tumor tissue, while the immunofluorescence staining only can show a very small region of the tumors. Sometimes, even the whole tumor is not very hypoxic, but you can still find a small hypoxic region inside.

2) Again, the data in Fig 6d is not enough to support the key point of this work. There are three control groups: no laser and no MPdot, laser only, and MPdot only. It is not surprising that only the treatment group (laser+MPdot) showed positive treatment effect, because only this group had the intact components for PDT, i.e., laser and photosensitizer. The key point of this work is that the authors have developed new photosensitizers for type-I PDT, which are better than type-II photosensitizers in the treatment of hypoxic tumors. So, at least Fig 6d should include a control group that is performed by using laser + type-II photosensitizer. If this control group does not work well for the treatment of the same 4T1 tumor model, then the value of the developed MPdot will be proved.

3) To enhance the targeting ability, MPdot was modified with c(RGDyK) peptide, and then photoacoustic imaging of the tumor over time was performed to monitor the tumor accumulation and retention of MPdot. In Fig 6c, it is hard to understand why only the images of one group are demonstrated. There should be at least two groups of photoacoustic images of MPdot with and without c(RGDyK) peptide. If not, the authors may doubt that perhaps c(RGDyK) peptide conjugation is not necessary.

4) Many endogenous molecules, such as hemoglobin, water and lipids can generate strong photoacoustic signals upon NIR laser excitation. It is not clear how the authors extract the signal of the MPdot from the total mixed signals. In the methods part ("In Vivo Photoacoustic Imaging"), no description about the signal unmixing can be found. It is not even clear which excitation wavelength was used to collect these images. The authors did not examine the photoacoustic spectrum of the MPdot (see next question), how did they know which excitation wavelength should be used for in vivo photoacoustic imaging?

5) The characterization of the photoacoustic properties of MPdot in solution is too simple, only one image is demonstrated in Figure S22. The MPdot only has weak fluorescence, so the biodistribution and tumor accumulation of the MPdot after i.v. injection only can rely on the in vivo photoacoustic imaging. Therefore, before in vivo photoacoustic imaging, the photoacoustic properties of MPdot in

solution should be carefully studied, e.g., the photoacoustic spectra of the MPdots and photoacoustic amplitudes at various concentrations. Only based on these information, the authors can correctly choose the excitation wavelengths for collecting the images and then unmix the photoacoustic signals.

Responses to Reviewers

Reviewer #1 (Remarks to the Author):

I think the manuscript can be published in its current version.

Response: We thank the reviewer for reviewing our revised manuscript and positive comment.

Reviewer #2 (Remarks to the Author):

The revised manuscript is indeed improved compared to the previous one. However, the strategy employed in this work may not possess the necessary novelty required for publication in Nature Communications. The electron transfer between the metal and ligand for type I photosensitization and metallo polymerization are well-established and widely used conventional strategies. Therefore, the manuscript would be more suitable for submission to specialized journals in the field.

Response: We thank the reviewer for the positive response to our improvement and further concern on the novelty. The ligand-to-metal charge transfer is a basic phenomenon in coordination complexes, which has great influence on the photophysical and photochemical properties. This electron transfer is not the main body and key topic of our work. We cite the literature to further theoretically support our results. The problem of type-I small molecular Ir(III) photosensitizers is the short and weak absorbance. The focus of our work is that we screened Type-I small molecular Ir(III) complexes and aimed to develop a **useful strategy** to prepare Type-I photosensitizers with much improved absorption properties and high-performance of ROS generation via molecular engineering, making the Ir-complexes more practical for therapy. The results can sufficiently support our view. The excellent therapeutic outcomes of the corresponding nanoparticles have been realized.

Figure R1. The absorption of example Ir(III) metallopolymer systems for photodynamic therapy and the wavelength of the irradiation laser marked with red arrow. Figures from reference literatures: (a). *Adv. Funct. Mater.* 2014, 24, 4823, Figure 5a. Copyright © 2014 WILEY - VCH Verlag GmbH & Co. KGaA, Weinheim; (b). *ACS Appl. Mater. Interfaces* 2017, 9, 28319, Figure 2e. Copyright © 2017 American Chemical Society; (c). *Chem. Sci.*, 2019, 10, 5085, Figure 1c. Copyright © The Royal Society of Chemistry 2019.

Secondly, very limited metallopolymer photosensitizers were indeed reported but showed major drawbacks of absorption as shown in **Figure R1**, or hypoxia intolerant type-II photochemistry pathway. Our strategy addressed absorption, ROS generation efficiency, and Type-I issues at one time, and exhibited generality, which is anticipated to find broad use in exploration of type-I photosensitizers based on donor-acceptor type narrow bandgap metallopolymer systems.

In short, the presence of some existing reports does not undermine the novelty of our work but on the contrary, lends strong support to it by making our research well-rooted and providing examples with flaws. The advantage of the donor-acceptor metallopolymer photosensitizers are obvious and we believe that the clear design guideline in this work will raise the attention in the scientific community. We thank you again for your valuable comments and consideration.

Reviewer #3 (Remarks to the Author):

The authors have added the data of hypoxia detection in tumor tissue with commercially available HpoxyprobeTM-1 Green Kit, which is helpful to support their conclusion. Here are the remaining issues to be addressed:

1) The hypoxia staining images by HpoxyprobeTM-1 Green Kit is shown in the response letter, but it seems that these images are not included in main text or supplementary information. Besides, I see the authors have performed photoacoustic imaging of the tumor tissues over time. I don't understand why they didn't characterize the tumor oxygenation state by photoacoustic imaging, which can be easily performed by analyzing the signals from the endogenous hemoglobin. This will be much better than the immunofluorescence staining method, because the PA imaging can demonstrate the hypoxia state of the whole tumor tissue, while the immunofluorescence staining only can show a very small region of the tumors. Sometimes, even the whole tumor is not very hypoxic, but you can still find a small hypoxic region inside.

Response: We thank the reviewer for positive comments on hypoxia detection to support our conclusion. We have included the hypoxia staining results in the updated version of SI. We highly value the reviewer's concerns about the precision of the immunofluorescence staining results to determine the hypoxic region in the tumor. Investigating hypoxia with the PAI technique via monitoring endogenous hemoglobin is good advice, and we have conducted the PA characterization on the Vevo LAZR system (FUJIFILM VisualSonics, Toronto, Canada), and the results have been presented in **Figure R2** and read the sO₂ in tumor and surface tissue on the Vevo LAZR software (refer to *Nat Methods* **12**, iii–v (2015); *ACS Nano* **2020**, 14, 2, 2063; *Adv. Healthcare Mater.* **2023**, 12, 2300110). Based on our new results shown in **Figure R2**, the hypoxia in tumor region was confirmed.

Figure R2. sO₂ and HbT in the epidermal area (Region1) and tumor area (Region 2) were measured using the Vevo LAZR imaging system to delineate the area of interest (40-dB gain, 10.00 mm depth, 12.08 mm width).

2) Again, the data in Fig 6d is not enough to support the key point of this work. There are three control groups: no laser and no MPdot, laser only, and MPdot only. It is not surprising that only the treatment group (laser+MPdot) showed positive treatment effect, because only this group had the intact components for PDT, i.e., laser and photosensitizer. The key point of this work is that the authors have developed new photosensitizers for type-I PDT, which are better than type-II photosensitizers in the treatment of hypoxic tumors. So, at least Fig 6d should include a control group that is performed by using laser + type-II photosensitizer. If this control group does not work well for the treatment of the same 4T1 tumor model, then the value of the developed MPdot will be proved.

Response: We sincerely thank the reviewer's concerns about this issue, and the advice to conduct a Type-II photosensitizer-based experiment to validate the necessity, which would make the strategy more logically sound and substantiated. The hypoxia-tolerance of Type-I materials over Type-II has been conventionally acknowledged (refer to *Coordination Chemistry Reviews*, **2022**, 452, 214306; *Adv. Mater.* **2021**, 33, 2103978; *Coordination Chemistry Reviews*, **2021**, 427, 213604). Meanwhile, the impeded therapy efficacy of Type-II photosensitizers in tumor due to hypoxia have been addressed in many works (refer to *Nat. Commun.* **2015**, 6(1), 8785; *J. Am. Chem. Soc.* **2015**, 137, 1539; *Chem. Sci.*, 2019, **10**, 5766).

We value the reviewer's advice and further supplemented experiments to substantiate this point. We synthesized a Type-II polymeric photosensitizer and completing in-solution and *in vivo* evaluation. The **p-BODIPY-F** polymer is chosen due to the similar absorption behavior around 680 nm to **Ir-P1**. Secondly, we confirmed its Type-II ROS generation with scavenger experiments, as shown in **Figure R3**. Thirdly, the *in vitro* and *in vivo* therapy parameters were tuned based on the former experimental conditions, according to the solution detection results.

Figure R3. (a). Chemical structure of **p-BODIPY-F** and absorption spectra of **PBdot** (10 ppm, 9.18 μM). ROS generation without or in the presence of scavengers. (b). No scavenger; (c). **Tiron** as O₂^{•-} scavenger; (d). **t-BuOH** as OH[•] scavenger. Experimental conditions: [DPBF] = 50 μM, [PBdot] = 10 ppm = 9.18 μM, [Tiron] = [t-BuOH] = 50 mM with a 680 nm laser (100 mW·cm⁻²) and recorded every 30 s.

Figure R4. (a). HeLa cell viability assay with varying concentrations of type-II Pdotr **p-BODIPY-F**: blue column, cells without laser treatment; green column, cells with laser treatment in hypoxia ($V\%O_2 = 1\%$) and orange column, cells with laser treatment in normoxia ($V\%O_2 = 21\%$), respectively. Experimental condition: **PBdotr** incubation duration, 20 hours; laser treatment, a 680 nm laser at a power of $200 \text{ mW}\cdot\text{cm}^{-2}$ for 20 mins (b). Tumor weight measured after dissection. (c). Mice photo and (d). Tumor photo for mice in each group. Experimental condition: Pdot administration, $100 \mu\text{L}$ of c(RGDyK)-modified **p-BODIPY-F** at a concentration of $400 \mu\text{g}/\text{mL}$; laser treatment, 680 nm laser with a power of $400 \text{ mW}\cdot\text{cm}^{-2}$ for 15 mins. ***, $P < 0.001$

The *in vitro* and *vivo* experiment results are listed in **Figure R4** and PBdot of **p-BODIPY-F** failed to kill the tumor cells efficiently *in vitro* under hypoxia condition ($1\% O_2$). It should be noted that **PBdotr** of **p-BODIPY-F** showed O_2 -concentration dependent behavior (**Figure R4(a)**) while **MPdot1r** and **MPdot2r** remained their therapeutic ability even under low level (1%) of oxygen (**Figure R5(a, b, d, e)**), (data are also supplemented in the revised manuscript and SI). It also should be noted that **PBdotc** of **p-BODIPY-F** failed to exhibit excellent therapeutic result at animal level. Possible main reason could be the poor photosensitization of Type-II photosensitizer in the presence of low level of O_2 . Your concern is highly valued, and consequently, we have added discussion in

the revised manuscript and added the results in **Figure R4** and **Figure R5** to the revised SI (**Figure S27**).

Figure R5. Cell viability and *in vivo* therapeutic results. Experimental conditions: HeLa cell was preincubated with **MPdot1r** or **MPdot2r** for 20 hours before 680 nm laser irradiation ($100 \text{ mW}\cdot\text{cm}^{-2}$) for 10 mins and animals was treated with **MPdot1c** or **MPdot2c** ($100 \mu\text{L}$, $400 \mu\text{g/mL}$) and laser (680 nm laser with a power of $300 \text{ mW}\cdot\text{cm}^{-2}$ for 10 mins). (a). **MPdot1r**, normoxia ($V\%_{\text{O}_2} = 21\%$); (b). **MPdot1r**, hypoxia ($V\%_{\text{O}_2} = 1\%$); (c). Tumor harvested after **MPdot1c** therapy; (d). **MPdot2r**, normoxia ($V\%_{\text{O}_2} = 21\%$); (e). **MPdot2r**, hypoxia ($V\%_{\text{O}_2} = 1\%$); (f). Tumor harvested after **MPdot2c** therapy.

3) To enhance the targeting ability, MPdot was modified with c(RGDyK) peptide, and then photoacoustic imaging of the tumor over time was performed to monitor the tumor accumulation and retention of MPdot. In Fig 6c, it is hard to understand why only the images of one group are demonstrated. There should be at least two groups of photoacoustic images of MPdot with and without c(RGDyK) peptide. If not, the authors may doubt that perhaps c(RGDyK) peptide conjugation is not necessary.

Response: Thank you for the advice. We have added images of two groups in revised manuscript according to your suggestion. Nanoparticles without peptide

coating can passively enrich in tumor due to EPR effect. We systematically compared the targeting ability of two MPdots with/without c(RGDyK) peptide, and the results are presented in **Figure R6**. The c(RGDyK) modification can further enhance the enrichment of nanoparticles in solid tumor. The time-dependent PA intensity *in vivo* results are shown below (also included in SI, **Figure S23**).

Figure R6. PA images of mice with different metallopolymeric nanoparticles. The pictures were taken at 0 and 12 hours after MPdot administration (400 pp, 100 μ L). (a₁, a₂). Control without MPdot; (b₁, b₂). **MPdot1** with c(RGDyK) modification; (c₁, c₂). **MPdot1** with no modification; (d₁, d₂). **MPdot2** with c(RGDyK) modification; (e₁, e₂). **MPdot2** with no modification. Experimental condition: 680 nm, PA-Model(single), 100% power, 40-dB gain, 40-MHz frequency. The intensity of the photoacoustic signal in the tumor region was determined by delineating the region of interest by the imaging system.

4) Many endogenous molecules, such as hemoglobin, water and lipids can generate strong photoacoustic signals upon NIR laser excitation. It is not clear how the authors extract the signal of the MPdot from the total mixed signals. In the methods part (“In Vivo Photoacoustic Imaging”), no description about the signal unmixing can be found. It is not even clear which excitation wavelength was used to collect these images. The authors did not examine the photoacoustic spectrum of the MPdot (see next question). how did they know which excitation wavelength should be used for in vivo photoacoustic imaging?

Response: We thank the reviewer for the comments and concerns on this issue.

The 680 nm laser was used since the two metallopolymers have high absorption in this region and PA intensity is highly related to optical absorption coefficient (refer to *Rev. Sci. Instrum.* **2007**, 78, 043102; *ACS Appl. Mater. Interfaces* **2018**, 10, 8, 7012). In former experiments, the codes for unmixing signals and our home-made system were old and have already been peer-reviewed in published work, and consequently not mentioned (refer to *Photoacoustics*, **2020**, 19, 100179; *ACS Appl. Mater. Interfaces* **2018**, 10, 8, 7012; *Sensors and Actuators B*, **2018**, 267, 403). In this revision, we performed the PA characterization on more advanced Vevo LAZR system (refer to *Nat Methods* **12**, iii–v (2015); *ACS Nano* **2020**, 14, 2, 2063; *Adv. Healthcare Mater.* **2023**, 12, 2300110). No unmixing process was involved. The parameters were set as listed in the **Table 1**. The intensity of the photoacoustic signal was determined by delineating the region of interest by the Vevo LAZR imaging system. sO₂ and HbT in the epidermal area and tumor area were measured using an imaging system to delineate the area of interest. The methods and results have been updated SI (refer to **Methods and Materials, Photoacoustic Effect in Solution, In Vivo Photoacoustic Imaging, Figure S22, Figure S23, and Figure S26**). The photoacoustic spectrum of the MPdots are shown in Figure R7 below.

Vevo LAZR system	Test	Model	Acquisition											
			Wavelength Range	Frame Rate	PA Gain	2D Gain	Depth	Width	Sensitivity	Resp Gate	Persistence	PA Acquisition	Wavelength [nm]	Correct Energy
	Materials in Solution	Single	680-970	5	40.0	29.0	10.0	12.08	High	Off	Low	Single	680	On
	Accumulation in vivo	Single	680-970	5	41.0	29.0	10.0	12.08	High	Off	Low	Single	680	On
	Oxygenation	Oxy-Hemo	680-970	-	40.0	29.0	10.0	12.08	High	Off	Low	sO ₂ /HbT	750/850	On
	Test	Model	Transmit			Display								
			Frequency [MHz]	2D Power [%]	PA Power [%]	Display Type	Display Map	Priority [%]	Threshold HbT	Brightness	Contrast	PA Brightness	PA contrast	
	Materials in Solution	Single	40	100	100	-	PA1	99	-	50	50	55	80	
	Accumulation in vivo	Single	40	100	100	-	PA1	99	-	50	50	59	80	
	Oxygenation	Oxy-Hemo	40	100	100	OxyZated	PA2	99	13	44	41	-	-	

Figure R7. Photoacoustic spectra for MPdots. (a, b) **MPdot1c**; (c,d) **MPdot2c**. Experimental condition: for photoacoustic spectra, the photoacoustic signal was acquired at 680, 685, 700, ..., 745, 750 nm (100 ppm **MPdot1c** or 50ppm **MPdot2c** in PBS, PA-Model(single), 100% power, 40-dB gain, 40-MHz frequency). For concentration-dependent photoacoustic spectra, the photoacoustic signal was acquired at 0, 6.25, 12.5, 25, and 50 ppm of **MPdot1c** or **MPdot2c** in PBS (680 nm, PA-Model(single), 100% power, 40-dB gain, 40-MHz frequency).

5) The characterization of the photoacoustic properties of MPdot in solution is too simple, only one image is demonstrated in Figure S22. The MPdot only has weak fluorescence, so the biodistribution and tumor accumulation of the MPdot after i.v. injection only can rely on the in vivo photoacoustic imaging. Therefore, before in vivo photoacoustic imaging, the photoacoustic properties of MPdot in solution should be carefully studied, e.g., the photoacoustic spectra of the MPdots and photoacoustic amplitudes at various concentrations. Only based on this information, the authors can correctly choose the excitation wavelengths for collecting the images and then unmix the photoacoustic signals.

Response: We sincerely thank the reviewer for the professional comments on

this issue. We have conducted the concentration dependent PA intensity spectra, as shown in **Figure R7** below. The results have been also added to SI.

Figure R7. Photoacoustic spectra for MPdots. (a, b) **MPdot1c**; (c,d) **MPdot2c**. Experimental condition: for photoacoustic spectra, the photoacoustic signal was acquired at 680, 685, 700, ..., 745, 750 nm (100 ppm **MPdot1c** or 50ppm **MPdot2c** in PBS, PA-Model(single), 100% power, 40-dB gain, 40-MHz frequency). For concentration-dependent photoacoustic spectra, the photoacoustic signal was acquired at 0, 6.25, 12.5, 25, and 50 ppm of **MPdot1c** or **MPdot2c** in PBS (680 nm, PA-Model(single), 100% power, 40-dB gain, 40-MHz frequency).

Reviewers' Comments:

Reviewer #2:

Remarks to the Author:

"The focus of our work is that we screened Type-I small molecular Ir(III) complexes and aimed to develop a useful strategy to prepare Type-I photosensitizers with much improved absorption properties and high-performance of ROS generation via molecular engineering, making the Ir-complexes more practical for therapy."

I understood that the present study was the discovery of long wavelength Ir(III) metallopolymer and its usefulness for PDT after the screening. However, the absorption wavelength did not reach the NIR region, and the ROS generation quantum yield is not quantitative and unclear in this manuscript.

"Our strategy addressed absorption, ROS generation efficiency, and Type-I issues at one time, and exhibited generality, which is anticipated to find broad use in exploration of type-I photosensitizers based on donor-acceptor type narrow bandgap metallopolymers."

If the authors emphasize the absorption intensity and the efficiency of ROS generation yield, it is necessary to analyze the reasons behind the excellent absorption and ROS generation efficiency exhibited by the metallo-polymer from a detailed process perspective. This analysis should be conducted from the point of view of the photo-absorption properties, charge separation generation yield, charge lifetime, and charge carrier mobility.

In my opinion, it seems that the novelty might not have reached to Nature Communications, but I will defer to the judgment of other referees.

Reviewer #3:

Remarks to the Author:

In the revised manuscript, the authors have synthesized a new Type-II polymeric photosensitizer (p-BODIPY-F) with similar absorption behavior around 680 nm to Ir-P1, and tested the in vitro and in vivo therapy parameters, which is helpful to demonstrate the advantage of the Type-I photosensitizer Ir-P1. Besides, the characterization data showing the in vitro photoacoustic properties of MPdots have also been added. The novelty is indeed a weak part of this paper. I think the authors should at least improve the writing of the introduction and discussion parts to better emphasize the key contribution of this work and try to better persuade the readers why this work represents as a universal strategy. In the current version, it seems that type-I metallopolymer photosensitizer has already been reported in the literature, while the key topic of this work is to improve some detailed parameters, including the absorption coefficient, absorption wavelength, and ROS production efficiency. This will leave an impression to the reviewers and the future readers that this work is just an incremental progress. In addition, the authors should carefully check the quality of the added data. For example, in Figure R2, the sO₂ in region 2 was measured to be 0%. Is this correct? It seems that no photoacoustic signal was detected from hemoglobin in region 2 and therefore the sO₂ could not be measured. In Figure R7 (b and d), why the x axis (0 – 60) was labeled as "wavelength (nm)"? The data of p-BODIPY-F are important but now totally put in the supporting information. Some of the key data can be moved to the main body.

REVIEWER COMMENTS

Reviewer #2 (Remarks to the Author):

“The focus of our work is that we screened Type-I small molecular Ir(III) complexes and aimed to develop a useful strategy to prepare Type-I photosensitizers with much improved absorption properties and high-performance of ROS generation via molecular engineering, making the Ir-complexes more practical for therapy.”

I understood that the present study was the discovery of long wavelength Ir(III) metallopolymer and its usefulness for PDT after the screening. However, the absorption wavelength did not reach the NIR region, and the ROS generation quantum yield is not quantitative and unclear in this manuscript.

Response: We thank the reviewer for commenting on this issue. The major contribution of our work aimed at exploring a method with generality to synthesize high-performance, intense long-wavelength absorbing, Type-I ROS-generating metallopolymer. Based on our results, the strategy works well in blue, green, and deep-red regions (670 nm, in the biological window). To further extend absorption to NIR regions, we have explored tuning the ratio of **Ir1** and **DPP** units to manipulate the absorption behavior of the resulting metallopolymer. **Ir-P44** and **Ir-P37**, in which the iridium unit took up 44.4% and 37.5% of total structure units, respectively, exhibited increased absorption in NIR regions. We evaluated the ROS generation (shown in **Figure R1** below). These metallopolymer maintained Type-I ROS-generating property with reasonable ROS generation efficiency under 808 nm irradiation. The Type-I ROS generation was confirmed by EPR measurement (**Figure R1**, e,f) and in vitro cell imaging. As shown in **Figure R2** and **R3**, the generation of $O_2^{\cdot-}$ was verified by the activation of **DHE** probe in living cells. We did not further conduct in vivo experiments since a strong photothermal effect appeared, which is an interference factor when we discuss Type-I PDT in vivo. Actually, the photothermal effect of various NIR materials is very common. In short, efficient Type-I ROS photosensitizers with NIR light activation could be achieved under our design guidelines. We have added some discussion about the NIR metallopolymer in the revised manuscript.

Figure R1. (a) The structure of metallopolymers; (b) normalized absorption spectra of metallopolymers; (c, d) **DPBF** for ROS detection in dichloromethane solutions for **Ir-P44** and **Ir-P37**. Experimental conditions: $[\text{DPBF}] = 50 \mu\text{M}$, $[\text{Ir-P44}] = [\text{Ir-P37}] = 14.0 \mu\text{M}$ with an 808 nm laser ($1 \text{ W}\cdot\text{cm}^{-2}$) and recorded every 30 s. (e, f) The EPR data was obtained from an aqueous solution ($\text{DMSO}:\text{H}_2\text{O} = 1:9$, v/v) containing **BMPO**. Experimental conditions: $[\text{MPdot44}] = [\text{MPdot37}] = 0.5 \text{ mM}$, $[\text{BMPO}] = 100 \text{ mM}$; irradiation: 808 nm laser ($1 \text{ W}\cdot\text{cm}^{-2}$).

Figure R2. ROS assay with **DHE**. Experimental conditions: [MPdot44r] = 50 μg/mL with preincubation for 20 hours, [DHE] = 5 μM with incubation for 30 mins, 808 nm laser irradiation ($1\text{W}\cdot\text{cm}^{-2}$) for 5 mins. (a₁-a₃). Laser-/MPdot44r-; (b₁-b₃). Laser+/MPdot44r-; (c₁-c₃). Laser-/MPdot44r+; (d₁-d₃). Laser+/MPdot44r+.

Figure R3. ROS assay with **DHE**. Experimental conditions: [MPdot37r] = 50 μg/mL with preincubation for 20 hours, [DHE] = 5 μM with incubation for 30 mins, 808 nm laser irradiation ($1\text{W}\cdot\text{cm}^{-2}$) for 5 mins. (a₁-a₃). Laser-/MPdot37r-; (b₁-b₃). Laser+/MPdot37r-; (c₁-c₃). Laser-/MPdot37r+; (d₁-d₃). Laser+/MPdot37r+.

Figure R4. Concentration-dependent photothermal effects of **MPdot44** (a) and **MPdot37** (b). Experimental condition: 808 nm laser, $1 \text{ W} \cdot \text{cm}^{-2}$, 250 μL in 96-well plate, and data recorded every 10 s.

We also thank the reviewer for the concern about the ROS generation quantum yield. The quantum yields were measured using Ce6 as reference (75% in water). The ROS generation quantum yields of the **Ir-P1** and **Ir-P2** are 33% and 37%, respectively. It is noted that the absorption at 660 nm of **Ir-P1** and **Ir-P2** are 4.4 and 3.3 times (calculated based on the molarity of repeating units) as high as **Ce6**. Therefore, the absolute quantities of ROS generated from **Ir-P1** and **Ir-P2** are higher than **Ce6**, which is compensated with high-intensity absorption. Additionally, metallopolymers **Ir-P1** and **Ir-P2** show much better photostability than that of **Ce6** (**Figure R5**).

Figure R5. Photostability of photosensitizers. (a). **Ce6**; (b). **MPdot1**; (c). **MPdot2**. Experimental condition: $[\text{Ce6}] = [\text{MPdot1}] = [\text{MPdot2}] = 2.8 \mu\text{M}$, irradiated with 660 nm laser ($300 \text{ mW} \cdot \text{cm}^{-2}$) and recorded every 1 min.

“Our strategy addressed absorption, ROS generation efficiency, and Type-I issues at one time, and exhibited generality, which is anticipated to find broad use in exploration of type-I photosensitizers based on donor-acceptor type narrow bandgap metallopolymers.”

If the authors emphasize the absorption intensity and the efficiency of ROS generation yield, it is necessary to analyze the reasons behind the excellent absorption and ROS generation efficiency exhibited by the metallo-polymer from a detailed process perspective. This analysis should be conducted from the point of view of the photo-absorption properties, charge separation generation yield, charge lifetime, and charge carrier mobility.

In my opinion, it seems that the novelty might not have reached to Nature Communications, but I will defer to the judgment of other referees.

Response: We thank the reviewer for suggesting these measurements and concern about the novelty. We would like to respond one by one below.

❖ Photo-absorption properties

Firstly, introducing a rigid segment to expand the conjugation would promote the absorption coefficients of the molecules (refer to *Energy Environ. Sci.*, **2014**, *7*, 4118; *Nat. Mater.*, **2016**, *15*, 746). Metallopolymerization, rather than developing iridium-based small molecules can naturally take advantage of this feature, and thus greatly enhance the photo-absorption properties and finally boot rapid ROS generation.

❖ Charge separation generation yield, and charge lifetime

Figure R6. Structure of materials used in the transient absorption spectra.

To analyze the charge lifetime and charge separation (CS) generation yield, we conduct transient absorption measurements by using 400 nm excitation.¹ A close look at the time profiles shown in **Figure R7** demonstrates the elongated lifetime brought by the charge transfer process. We also derived charge lifetime from their photobleaching (PB) kinetics through exponential decay fitting

as 1684.6, 1574.0, 3410.6, and 1207.6 ps, for **MPdot1**, **MPdot2**, **MPdotF8**, and **MPdotTBT**, respectively.² In comparison, way shorter lifetimes could be derived as 614.8, 323.2, 13.4, and 402.7 ps for references **PdotDPP**(nanoparticle of **PPy-DPP**), **NP-QZL** (nanoparticle of **QZL-DPP**), **PdotF8**, and **PdotTBT**. The results above indicate that combination with Ir(III) significantly extended charge lifetimes of polymers.

In deriving their charge separation generation yield, we first traced the absorption features from the radical cation signals. For example, in **Figure R7** (a₁), by tracing the broad absorption around 750 nm that belongs to DPP⁺ radical cation³, we derived the charge separation lifetime of this system as 0.12 ps. A comparison with the competition process kinetics we derived from the reference **PdotDPP**, we get the CS yield as 99.1% for sample **MPdot1** and 97.9% for sample **MPdot2**.⁴ Due to the overlap with the strong energy transfer process as shown in **Figures R7** (e₁, e₂, f₁ and f₂) and (g₁, g₂, h₁ and h₂), we failed to derive the charge separation kinetic and accurate yield for samples **MPdotF8** and **MPdotTBT**.⁵ But it is believed that the elongated lifetime will definitely have a positive impact on ROS efficiency.

Figure R7. Differential absorption spectra were obtained upon femtosecond pump probe experiments with time delays between 1 and 5000 ps. Time absorption profiles and corresponding fits at different wavelengths. (a₁, a₂). PdotDPP for PPy-DPP; (b₁, b₂). MPdot1 for Ir-P1; (c₁, c₂). Nanoparticle (NP-QZL) for QZL-DPP; (d₁, d₂). MPdot2 for Ir-P2; (e₁, e₂). PdotF8 for PPy-F8; (f₁, f₂). MPdotF8 for Ir-PF8; (g₁, g₂). PdotTBT for PPy-TBT; (h₁, h₂). MPdotTBT for Ir-PTBT.

Reference:

[1] The femtosecond transient absorption (TA) spectra of different samples were taken using an Ultrafast System HELIOS TA spectrometer. The laser source was from the Coherent Astrella-1K-F Ultrafast Ti: Sapphire Amplifier (800nm, 1 kHz, <100 fs). The broadband probe pulses (450-775 nm) were generated by focusing a small portion of the fundamental 800 nm laser pulses into a Al₂O₃ plate. The 400-nm pump pulses were obtained through doubling the fundamental 800 nm pulses with a BBO crystal.

[2] In the time-profile fittings, we used following formula:

$$y = A + B_1 \exp\left(\frac{-t}{\tau_1}\right) + B_2 \exp\left(\frac{-t}{\tau_2}\right) + B_3 \exp\left(\frac{-t}{\tau_3}\right) \quad (1)$$

where B are relative amplitudes, and τ_1 , τ_2 and τ_3 represent the lifetimes of different photophysical pathway. In our case, the charge recombination is the last process and derived as τ_3 .

[3] *Photochem. Photobiol. Sci.*, **2010**, *9*, 1055.

- 【4】 In evaluating the kinetics of the competition process, we used the following formula considering the lifetime and relative amplitude of all the competition processes from the reference sample.

$$\tau = \frac{\sum B_i \tau_i}{\sum B_i} \quad (2)$$

The charge separation yield is estimated by considering the rate of charge separation and all the competition processes without charge separation. See table 1 for details. Note that the limited measurement resolution may cause the B_1 value to be smaller than the actual case, leading to high CS yield estimates. The charge separation lifetime is derived as 0.19 ps for **MPdot2**.

- 【5】 The energy transfer process is considered to happen from in polymer chain. The lifetime of this energy transfer process is derived from the rising edge of the PB signals as 5.2 ps for sample **MPdotF8** and 1.5 ps for sample **MPdotTBT**.

Table 1 Lifetime table of the competitive processes considered in **PdotDPP** and **NP-QZL**.

	B_1	τ_1 / ps	B_2	τ_2 / ps
PdotDPP	0.017	1.7	0.012	29.9
NP-QZL	0.010	1.8	0.005	22.7

❖ Charge carrier mobility

For charge carrier mobility, we would like to discuss with the reviewer and seek more professional advice. For inorganic semiconductors with compact crystalline structure such as quantum dots, the charge carrier mobility is very important for the charge transport to particle surface. Different from inorganic nanoparticles, the structure of polymer dots is loose (amorphous particles formed from aggregation of polymer chains). Because solvent molecules and oxygen can freely interact with charge in the loose particles, it is relatively difficult to measure the charge carrier mobility in water solution. Another way to measure charge carrier mobility is using thin film of these polymers, but the results highly depend on many factors, such as packing of polymer chains, crystallinity and thickness of polymer film, transporting layers, nature of the semiconductor–dielectric interface, etc. The results may not be suitable for explaining photochemical phenomena of polymer dots in water solution. Anyway, it is believed that the elongated charge lifetime will definitely have a positive impact on ROS efficiency.

We also thank the reviewer for the concern about the novelty. In the last “response to reviewers”, we cited three metallopolymers and discussed their drawbacks, which might cause some misunderstanding. Actually, none of them are type-I photosensitizers. To avoid misunderstanding by readers, we have also revised some confusing parts in the introduction and commented in a more direct way. The major contribution of our work aimed at exploring a general method for the synthesis of high-performance, intense long-wavelength absorbing, Type-I ROS-generating metallopolymers. Based on our results, the strategy shows good universality: it works well for metallopolymers in blue, green, deep-red, and NIR regions; high-performance type-I ROS generation could be achieved in metallopolymers where the iridium atom is covalently coupled to the polymer backbone, or directly included in the polymer backbones. The metallopolymer strategy is anticipated to provide general design guidelines for Type-I photosensitizers by judicious selection of metal complexes and organic building blocks. We sincerely thank the reviewer for valuable comments and many professional suggestions.

Reviewer #3 (Remarks to the Author):

In the revised manuscript, the authors have synthesized a new Type-II polymeric photosensitizer (p-BODIPY-F) with similar absorption behavior around 680 nm to Ir-P1, and tested the in vitro and in vivo therapy parameters, which is helpful to demonstrate the advantage of the Type-I photosensitizer Ir-P1. Besides, the characterization data showing the in vitro photoacoustic properties of MPdots have also been added. The novelty is indeed a weak part of this paper. I think the authors should at least improve the writing of the introduction and discussion parts to better emphasize the key contribution of this work and try to better persuade the readers why this work represents as a universal strategy. In the current version, it seems that type-I metallopolymer photosensitizer has already been reported in the literature, while the key topic of this work is to improve some detailed parameters, including the absorption coefficient, absorption wavelength, and ROS production efficiency. This will leave an impression to the reviewers and the future readers that this work is just an incremental progress.

Response: We thank the reviewer for the positive comment about the efforts we made and the concern about the novelty. In the last “response to reviewers”, we cited three metallopolymers and discussed their drawbacks, which might cause some misunderstanding. Actually, none of them are type-I photosensitizers. We have also revised the abstract, making it more concise and easier for general readers. We have also modified some confusing parts in the introduction and commented in a more direct way. The major contribution of our work aimed at exploring a method with generality to synthesize high-performance, intense long-wavelength absorbing, Type-I ROS-generating metallopolymers. Based on our results, the strategy works well for metallopolymers with absorptions in blue, green, and deep-red regions. To further validate the generality, we extended absorption of metallopolymers to NIR regions, which could generate Type-I ROS under 808 nm laser irradiation, as shown in **Figure R8** below.

Based on our results, the strategy shows good universality: it works well for metallopolymers in blue, green, deep-red, and NIR regions; high-performance type-I ROS generation could be achieved in metallopolymers where the iridium atom is covalently coupled to the polymer backbone, or directly included in the polymer backbones. This strategy may have prompt effects on practice-oriented research by developing more metallopolymers via very simple chemical synthesis. We have revised the confusing expression in the introduction about commenting on polymer’s advantages and previous work.

Figure R8. (a). The structure of **Ir-PF8**, **Ir-PTBT**, **Ir-P44**, and **Ir-P37**. The EPR data (b-e) was obtained from an aqueous solution (DMSO:H₂O = 1:9, v/v) containing **BMPO**. Experimental conditions: [MPdotF8] = [MPdotTBT] = [MPdot37] = [MPdot44] = 0.5 mM, [BMPO] = 100 mM; irradiation: white LED light (50 mW·cm⁻²) and 808 nm laser (1 W·cm⁻²). The absorption spectra of **DPBF** in dichloromethane solutions. (f). normalized absorption for **Ir-P1**, **Ir-P44**, and **Ir-P37**; (g). **Ir-P44**; (h). **Ir-P37**. Experimental conditions: [DPBF] = 50 μM, [Ir-P44] = [Ir-P37] = 14.0 μM with an 808 nm laser (1 W·cm⁻²) and recorded every 30 s.

Reviewer: In addition, the authors should carefully check the quality of the added data. For example, in Figure R2, the sO₂ in region 2 was measured to be 0%. Is this correct? It seems that no photoacoustic signal was detected from hemoglobin in region 2 and therefore the sO₂ could not be measured.

Response: We thank the reviewer for this careful reviewing. It was caused by the high HbT threshold. When we use a photoacoustic imaging system to detect the blood oxygen content of tumors, in order to truly reflect the blood oxygen situation in the tumor and epidermal tissues, we adjust the parameter HbT threshold appropriately to avoid background signal interference. As

shown in the supplementary data, when the HbT threshold is 20, the background interference is 0.000%. When this threshold is adjusted to 6, interference signals will appear in the blank background above. However, even so, under different parameters, we can still see the ROI2 region, which means the blood oxygen content inside the tumor is extremely low. This data can effectively indicate that the tumor is in a state of hypoxia. To avoid misunderstanding, we have updated new figures in supporting information (threshold of HbT = 6).

Figure R9. sO₂ value read with different HbT thresholds. (a). HbT threshold = 20; (b). HbT threshold = 6.

Reviewer: In Figure R7 (b and d), why the x axis (0 – 60) was labeled as “wavelength (nm)”’? The data of p-BODIPY-F are important but now totally put in the supporting information. Some of the key data can be moved to the main body.

Response: We thank the reviewer for the careful reviewing. We have corrected the label in new version (also shown in Figure R10 below). We have added discussion about the tumor weights to the main body.

Figure R10. Photoacoustic spectra for MPdots. (a, b) **MPdot1c**; (c,d) **MPdot2c**. Experimental condition: for photoacoustic spectra, the photoacoustic signal was acquired at 680, 685, 700, ..., 745, 750 nm (100 ppm **MPdot1c** or 50ppm **MPdot2c** in PBS, PA-Model(single), 100% power, 40-dB gain, 40-MHz frequency). For concentration-dependent photoacoustic spectra, the photoacoustic signal was acquired at 0, 6.25, 12.5, 25, and 50 ppm of **MPdot1c** or **MPdot2c** in PBS (680 nm, PA-Model(single), 100% power, 40-dB gain, 40-MHz frequency).

Reviewers' Comments:

Reviewer #2:

Remarks to the Author:

This manuscript has been significantly improved. All of my concerns have been addressed. The current version of the manuscript will provide valuable information for readers. I recommend accepting the manuscript for publication in Nature Communications.

Reviewer #3:

Remarks to the Author:

The authors have made great effort to revise the manuscript. I think the manuscript is now suitable for publication.